# LEVERAGING DIFFUSION TRANSFORMERS FOR ROBUST STOCK FACTOR AUGMENTATION IN FINANCIAL MARKETS

## ABSTRACT

Data scarcity poses a significant challenge in training machine learning models for stock forecasting, often leading to low signal-to-noise ratio (SNR) and data homogeneity that degrade model performance. To address these issues, we introduce DiffsFormer, a novel approach utilizing artificial intelligence-generated samples (AIGS) with a Transformer-based Diffusion Model. Initially trained on a large-scale source domain with conditional guidance to capture global joint distribution, DiffsFormer augments training by editing existing samples for specific downstream tasks, allowing control over the deviation of generated data from the target domain. We evaluate DiffsFormer on two datasets using eight commonly used machine learning models, achieving relative improvements of 7.3% and 22.1% in excess return, respectively. Extensive experiments provide insights into DiffsFormer's functionality and its components, illustrating their roles in mitigating data scarcity and enhancing model performance.

## 1 INTRODUCTION

Accurate stock forecasting plays a crucial role in effective asset management and investment strategies (Zou et al., 2022). Its objective is to predict future stock behavior (*e.g.,* return ratios or prices) by analyzing relevant historical factors. Previous research (Zhang et al., 2017b; Feng et al., 2019; Xu et al., 2021) has explored various machine learning techniques; however, achieving desirable performance with these methods often requires an ample supply of high-quality data. The challenges posed by high random and homogeneous data make it difficult to meet the requirements for data quality, resulting in elevated forecasting errors and increased uncertainty. Figure 1 demonstrates the significance of addressing the data scarcity issue. As demonstrated, when this challenge is mitigated, the model exhibits a progressive and substantial excess return (§2 Eq.(3)). This improvement highlights the potential performance gains achievable through effective data augmentation strategies.

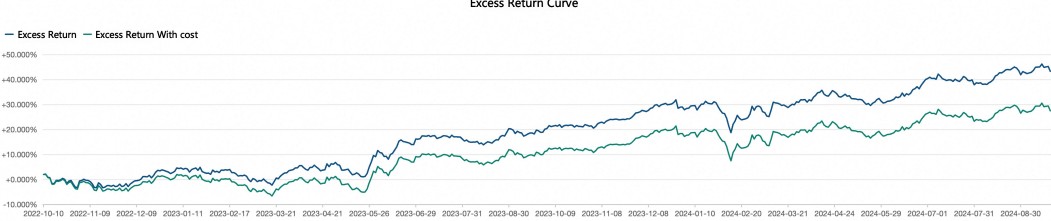

Figure 1: The cumulative excess return (§2 Eq.(3)) curve of our system on CSI300 index. Prior to 2023-04, the online model was the Transformer. Subsequently, DiffsFormer was deployed.

**Stock forecasting focuses on predicting (excess) return ratio with stock factors such as Open, Close, High and Low prices.** Data scarcity in the task can be delineated through two primary dimensions: *signal-to-noise ratio* (SNR, §2 Eq.(1)) and *data homogeneity* Firstly, we delve into the relationship between stock factors and the return ratio to elucidate insights regarding SNR. As illustrated in Figure 2a, the Pearson correlation coefficients between stock factors and the return ratio indicate a weak

correlation (with absolute values less than 0.03), which suggests a low SNR for these factors. This weak correlation is frequently attributed to randomness and non-stationary speculative behaviors in the market. Secondly, we assess the behavior of stocks within industry sectors to highlight the implications of data homogeneity. Our findings reveal that stocks within the same industry sector tend to exhibit similar behavior, as demonstrated in Figure 2b. The different colors in each bar represent various sectors, and the height of the color bar indicates the total number of stocks in specific sector facing price drops. The presence of substantial color blocks for specific sectors in certain years (e.g., larger blocks of blue, green and yellow in some years) suggests that when a sector is affected, it often impacts multiple stocks in that sector simultaneously. Consequently, this phenomenon of homogeneity diminishes the availability of stocks with unique informational characteristics. Such data scarcity presents inherent challenges, leading to the risk of overfitting, wherein models may learn shortcuts and spurious correlations, thereby adversely affecting their predictive performance. The limited availability of data constitutes a considerable obstacle to achieving effective generalization between training and testing datasets, ultimately compromising overall model performance.

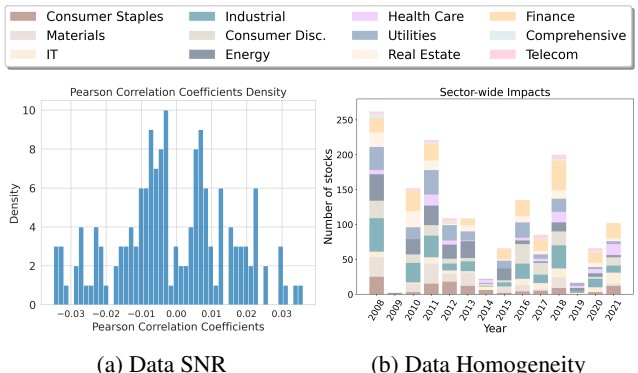

Drawing inspiration from the successful applications of Diffusion Models (DMs) in sequence generation (Tashiro et al., 2021; Rasul et al., 2021; Chen et al., 2020; Bilos et al., 2023; Alcaraz & Strodthoff, 2023), we propose a novel **Diff**usion Model designed to generate **s**tock factors using a Trans**former** architecture, referred to as DiffsFormer. Applying Diffusion Models (DMs) to factor augmentation in stock forecasting presents significant challenges. These challenges are twofold: (1) Unlike traditional DM applications, the stock forecasting context requires corresponding labels for the generated factors. (2) The inherent scarcity of financial data can hinder the generalization capabilities of DMs, potentially leading to overfitting on easily modeled patterns rather than capturing true market dynamics. To address these challenges, we have developed novel mechanisms that equip Diffusion Models with the capability to generate corresponding labels and mitigate overfitting issues.

(a) Data SNR

(b) Data Homogeneity

Figure 2: (a) Pearson Correlation Coefficients between return ratio and stock factors are low. (b) Number of stocks experiencing significant price drops in each sector.

In §3.1, we present the *knowledge transfer with edit mechanism*. Our proposed framework incorporates transfer learning to distill valuable knowledge and information from stocks in larger markets. DM with a diffusion step denoted as $T$ is first trained on a large source domain to **overcome the data scarcity nature**. During generation, rather than sampling from pure gaussian, we perturb data points from the target domain, and subsequently denoise to obtain new data points **with the same label** that resides within the target domain. Note that as the financial data is noisy, we restrict the perturb step to a small value $T' \ll T$, which we refer to as the editing step. On top of that, it is unnecessary to optimize the DM for $t > T'$ since they are never used during sampling. On top of that, in §3.2 we present the *time efficiency optimization* without affecting correctness.

In §3.3, we introduce the conditionings adopted for DM. Inspired by classifier-free guidance (Ho & Salimans, 2022), we equip DM with the capability to capture label and sector information which contributes to the **alignment of the generated feature and the original label and sector**. As the label for our task is continuous rather than discrete, we term our flexible conditional factor generation process as *predictor-free guidance*. In §3.4, we discover that the diffusion model overfits to some easily fitted patterns, hence we utilize the training loss as a proxy variable and introduce stronger noise to data points associated with lower training loss. This loss-guided noise addition mechanism aims to mitigate the volatility of the model by **addressing the overfitting issues** linked with easily fitted points, as opposed to employing uniform noise addition.

In summary, the contributions are as follows:

- We reveal the importance of data augmentation in the context of stock forecasting and explore the use of diffusion stock transformer (DiffsFormer for short) to address data scarcity.

- The framework integrates transfer learning to leverage knowledge from other markets, alleviating the difficulty of training DMs on sparse data. Additionally, the edit mechanism could obtain new features with original label with optimized efficiency, enabling training of the downstream forecasting task. For better alignment of the feature and the original label, we propose to employ excess return as the conditioning to enhance the relationship between them. A flexible predictor-free guidance approach is integrated as excess return is continuous rather than discrete.
- We verify the effectiveness of DiffsFormer augmented training in CSI300 and CSI800 with nine commonly used machine learning models.

## 2 BACKGROUND

In the ever-evolving landscape of financial markets, performance evaluation of a portfolio provides insights into investment strategies and helps in making informed decisions. With this in mind, in this section, we will introduce fundamental concepts and widely accepted evaluation methodologies crucial for assessing the performance and accuracy of stock price forecasting models.

**Stock Factors.** Factors are attributes of a stock that are identified as potential drivers of return.

**Signal-to-noise ratio (SNR).** Signal-to-noise ratio means the ratio of the signal power to the noise power. Generally, Data $X$ could be expressed as $S + N$, where $S$ is the signal variable, and $N$ is a random variable having an expected value equal to zero. Signal's power equals its mean-squared value, and the zero mean of the noise makes its power equal to its variance $\sigma^2$ (Johnson, 2006):

$$\text{SNR} = \frac{\mathbb{E}[S^2]}{\sigma^2} \tag{1}$$

**Return Ratio (RR).** The primary objective of stock forecasting is to achieve substantial profits. Previous study (Zou et al., 2022) treat RR as a metric to measure the model performance. RR serves as a crucial indicator to assess the success of stock forecasting models in achieving profitable investment outcomes. Following this setting, we define return ratio as:

$$\text{RR}(i) = \frac{P_{close}^{t+i} - P_{close}^{t}}{P_{close}^{t}}, \tag{2}$$

where $t$ is the current time, and $i$ denotes the time interval in days. $P_{close}^{t}$ denotes the current close price of the stock, and $P_{close}^{t+i}$ represents the close price of the same stock after $i$ days. Here, we calculate the return ratio on a daily basis, and often set $i$ to be 5.

**Excess Return.** Sometimes people care about how much a portfolio outperforms or underperforms a chosen benchmark index rather than the return itself. The excess return over an index is a measure used to evaluate the performance of an investment portfolio compared to a benchmark index (*e.g.,* CSI300 or CSI800 index). The formula for excess return is simple:

$$\text{Excess Return} = \text{Portfolio Return Ratio - Benchmark Return Ratio.} \tag{3}$$

**Information Coefficient (IC) and Rank information Coefficient (RankIC).** IC and RankIC (Lin et al., 2021; Li et al., 2019) are commonly used in finance and machine learning contexts to assess the effectiveness of predictive models. IC measures the Pearson correlation between predictions and actual labels, while Rank IC is concerned with Spearman's rank correlation between the two:

$$\text{IC} = \frac{\text{cov}(V_p, V_a)}{\sigma(V_p)\sigma(V_a)}, \quad \text{RankIC} = \frac{\text{cov}(\text{Rank}(V_p), \text{Rank}(V_a))}{\sigma(\text{Rank}(V_p))\sigma(\text{Rank}(V_a))}, \tag{4}$$

where $V_p$ and $V_a$ represent the vectors of predicted and actual values, respectively.

**Weighted IC.** In financial markets, especially where going short is banned, accurate modeling of tail stocks has little contribution to excess return compared to that of top stocks. Hence we introduce to apply an exponentially decayed weight on IC/RankIC to better characterize the correlation between the prediction and label on top stocks:

$$\text{WeightedIC} = \frac{\Sigma_{i=1}^{n}\omega_i(V_{p_i} - \overline{V_{p_\omega}})(V_{a_i} - \overline{V_{a_\omega}})}{\sqrt{\Sigma_{i=1}^{n}\omega_i(V_{p_i} - \overline{V_{p_\omega}})^2}\sqrt{\Sigma_{i=1}^{n}\omega_i(V_{a_i} - \overline{V_{a_\omega}})^2}}, \tag{5}$$

where $\omega_{i+1} = 0.99 * \omega_i$. $\overline{V_{p_\omega}}$ and $\overline{V_{a_\omega}}$ denotes the weighted average of vectors.

## 3 METHODOLOGY

The stock forecasting task is challenging primarily because of the scarcity of data. To harness the full potential of machine learning models, a sufficient amount of high-quality data is crucial. However, obtaining such high-quality stock data for a specific target domain is rare and can often be restricted as commercial secrets. In this work, we utilize the power of DM and introduce a novel approach, DiffsFormer. It generates additional data points and facilitates factor augmentation, enabling us to forecast the likely RR of real-world stocks despite data scarcity.

### 3.1 DIFFUSION-BASED DATA AUGMENTATION

Following (Ho et al., 2020; Nichol et al., 2021), DiffsFormer contains diffusion and denoising processes like most of the DMs do. The diffusion process parameterizes a Markov chain that progressively introduces noise to the factors until reaching a state of pure noise (Ho et al., 2020). Subsequently, during the denoising process, the model aims to restore the original data by predicting the noise generated through the diffusion process. This characteristic enables us to edit and augment sequential data. In this study, as shown in Figure 3, we look back 8 days and organize recent stock factors as a sequence, leveraging DMs based on transformer architectures (Peebles & Xie, 2022; Tashiro et al., 2021) to do factor augmentation. We expect that by incorporating augmented factors, our proposed model will exhibit enhanced resilience to data scarcity in the field of stock forecasting. Detailed explanation of denoising diffusion probabilistic model is shown in Appendix B.

**Diffusion process.** In stock forecasting, the input data $X \in \mathbb{R}^{n \times d \times k}$ consists of $n$ real stocks along with their recent $k$-day historical factors, for which $d$ is the factor dimension. We treat each stock $x$ (*i.e.,* a row of $X$) as $\mathbf{x}_0$ sampled from $q(\mathbf{x}_0)$, and add random noise to perform a transition according to equation 11. Thanks to the reparameterization trick (Ho et al., 2020), we can obtain the conditional distribution $q(\mathbf{x}_t|\mathbf{x}_0)$ for each stock (Wang et al., 2023; Tashiro et al., 2021):

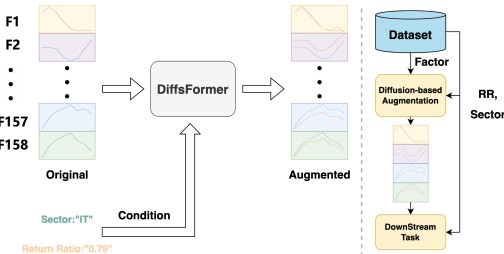

Figure 3: Sketch of DiffsFormer. F refers to "factors", such as the open/close/lowest/highest prices of stocks during a period.

$$q(\mathbf{x}_t|\mathbf{x}_0) = \mathcal{N}(\mathbf{x}_t; \sqrt{\overline{\alpha}_t}\mathbf{x}_0, (1-\overline{\alpha}_t)\boldsymbol{I}), \quad (6)$$

where $\overline{\alpha}_t = \prod_{i=1}^{t} \alpha_i$ and $\alpha_t = 1 - \beta_t$. Then, $\mathbf{x}_t$ is approximated as $\mathbf{x}_t = \sqrt{\overline{\alpha}_t}\mathbf{x}_0 + (\sqrt{1-\overline{\alpha}_t})\epsilon$ where $\epsilon \sim \mathcal{N}(0, \boldsymbol{I})$. $\alpha_i$ is related to the total diffusion step $T$.

**Denoising process.** During the denoising process, we subtract noise from $\mathbf{x}_t$ to recover the corresponding $\hat{\mathbf{x}}_0 \sim q(\mathbf{x}_0)$. Furthermore, we parameterize $p_\theta(\mathbf{x}_{t-1}|\mathbf{x}_t)$ through a neural network to estimate $q(\mathbf{x}_{t-1}|\mathbf{x}_t, \mathbf{x}_0)$. Specifically, we have $p_\theta(\mathbf{x}_{t-1}|\mathbf{x}_t) = \mathcal{N}(\mathbf{x}_{t-1}; \mu_\theta(\mathbf{x}_t, t), \boldsymbol{\Sigma}_q(t)\boldsymbol{I})$ with:

$$\mu_\theta(\mathbf{x}_t, t) = \frac{1}{\sqrt{\alpha_t}}(\mathbf{x} - \frac{\beta_t}{\sqrt{1-\overline{\alpha}_t}}\epsilon_\theta(\mathbf{x}_t, t))$$

$$\boldsymbol{\Sigma}_q(t) = \frac{(1-\overline{\alpha}_{t-1})\beta_t}{1-\overline{\alpha}_t}, \quad (7)$$

where $\epsilon_\theta(\mathbf{x}_t, t)$ is the trainable noise term to predict $\epsilon$ in the diffusion process.

**Objective.** The overall learning objective is to minimize the error in estimating $\epsilon$ with $\epsilon_\theta(\mathbf{x}_t, t)$ (Nichol et al., 2021). Formally, we aim to solve the following optimization problem:

$$\mathcal{L}_{da} = \min_\theta \mathbb{E}_{\mathbf{x}_0 \sim q(\mathbf{x}_0), \epsilon \sim \mathcal{N}(0,\mathbf{I}), t \sim \text{Uniform}} ||\epsilon - \epsilon_\theta(\mathbf{x}_t, t)||_2^2$$

$$s.t. \ \mathbf{x}_t = \sqrt{\overline{\alpha}_t}\mathbf{x}_0 + (\sqrt{1-\overline{\alpha}_t})\epsilon. \quad (8)$$

**Inference acceleration.** In Denoising Diffusion Probability Models, the lack of parallelism during the transition of DMs leads to slow inference. To tackle this problem, Denoising Diffusion Implicit Models (DDIM) (Song et al., 2020) accelerates samplings by modifying the forward process as:

$$q_\sigma(\mathbf{x}_{1:T} \mid \mathbf{x}_0) = q_\sigma(\mathbf{x}_T \mid \mathbf{x}_0) \prod_{t=2}^{T} q_\sigma(\mathbf{x}_{t-1} \mid \mathbf{x}_t, \mathbf{x}_0), \quad (9)$$

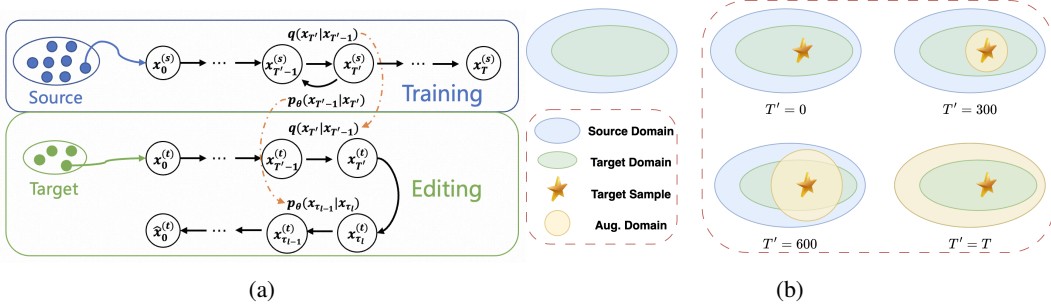

Figure 4: (a) The training and the editing topology. (b) Illustration of the editing step $T'$.

where $q_\sigma \left( \mathbf{x}_{t-1} \mid \mathbf{x}_t, \mathbf{x}_0 \right)$ is parameterized by $\sigma$ which means the magnitude of the stochastic process. When setting $\sigma_t = \mathbf{\Sigma}_q(t)$ for all steps, the forward process collapses to Markovian and the denosing process becomes the same as shown in Eq.(7). Specifically, when setting $\sigma_t = 0$, the corresponding denosing process becomes deterministic and thus sampling could be accelerated along the deterministic path. Technically, we follow the deterministic sampling design and create $\{\tau_i\}, \{i = 1 \cdots l\}$ as a sub-sequence of $\{t = 1, 2, \cdots, T\}$, where $l$ is the length of the sub-sequence. The denoising process can now be completed in just $l \ll T$ steps armed with DDIM sampling.

**Factor editing with transfer learning.** To alleviate data homogeneity issue, we augment the raw factors in target domain by going through a noising-denoisng process. Instead of generating synthetic factors from pure noise which hardly ensures data fidelity, we adopt a different approach by editing the original factors rather than generating entirely new ones. Moreover, due to the intrinsic low SNR nature of the factors, we design a transfer learning framework to distill new knowledge and information into edited data from a larger, different domain. Concretely, DiffsFormer of diffusion step $T$ is first trained on the source domain $\boldsymbol{X}^{(s)}$. During the inference process, we begin with a data point in the target domain $\mathbf{x}_0^{(t)}$, corrupt it for $T' \ll T$ steps to get a seed point: $\mathbf{x}_0^{(t)} \rightarrow \mathbf{x}_1^{(t)} \rightarrow \cdots \rightarrow \mathbf{x}_{T'}^{(t)}$. Then, we reverse the process from the seed to obtain a new data point $\mathbf{x}_{T'}^{(t)}$ in the target domain: $\mathbf{x}_{T'}^{(t)} \rightarrow \hat{\mathbf{x}}_{T'-1}^{(t)} \rightarrow \cdots \rightarrow \hat{\mathbf{x}}_0^{(t)}$. In our work, CSI300 and CSI800 are target domains (evaluation dataset), for which CSIS serves as the source domain. CSI 300 comprise the largest 300 stocks in the A-share market; CSI 800 adds some stocks to CSI300 with smaller size; CSIS means all stocks in the A-share market. Hence both of the target domains are a subset of the source domain, and this procedure distills new knowledge and information and enhances the data heterogeneity. Moreover, since the inference process starts from the seed, we can successfully edit existing samples. As illustrated in Figure 4b, $T'$ can control the strength of knowledge distillation: a larger $T'$ makes the generated data resemble the feature distribution of the source domain more closely, while a smaller $T'$ makes the generated data closer to the target domain data $\mathbf{x}_0^{(t)}$. We term $T'$ as the editing step. By doing so, we improve the fidelity of the generated data, avoiding creating data from pure noise. An illustration of the process is shown in Figure 4a. The detailed algorithms for training and inference are shown in Algorithms 1 and 2, respectively.

**The relationship between SDEdit (Meng et al., 2022).** SDEdit is a prestigous work in image edit domain, and have something in common with our edit mechanism: SDE serves as the theoretical support (SDE) for both of the problems, and the perturbing and reverse process looks alike. However, our approach differs in: SDEdit aims to generate both faithful and realistic image given input guidance image; while we expect diffsformer to: (1) be free from generation problems and (2) keep label unchanged. By training diffusion model in source domain and starting from seed sample in target domain during inference, we generates new sample with the same label with seed, aggregating information from the target domain, whose strength could be controlled by the editing step $T'$.

## 3.2 TIME EFFICIENCY IMPROVEMENT

From previous analysis in §3.1, it is obvious to see that there is no need to optimize $\epsilon_\theta(\mathbf{x}_t, t)$ for $t > T'$ under our transfer learning framework. Since DMs are time-consuming, we develop a trick to speed up the training of the framework. Concretely, we initialize $\alpha$ and $\beta$ with total diffusion steps $T$ to ensure correctness; however, we sample training step $t$ from $Uniform\{1, 2, \cdots, T'\}$ instead of $Uniform\{1, 2, \cdots, T\}$: compared to traditional DM, the probability of sampling useful steps

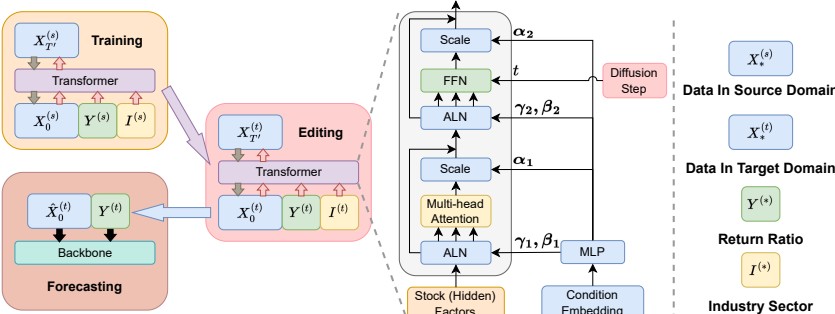

Figure 5: DiffsFormer overview. Y denotes label and I denotes industry sector. DiffsFormer incorporates transfer learning and conditional guidance to ensure improved model performance. Details for transformer architecture (Peebles & Xie, 2022) are introduced in Appendix C.10.

that are smaller than $T'$ is increased. The loss curves with maximum sampling steps within the set $\{100, 300, 500, 700, 1000\}$ are elucidated in Appendix C.10. We discover that with the decrease of sampling steps, DMs embrace with a more sharp loss curve, which means they can converge faster.

### 3.3 CONDITIONAL DIFFUSION AUGMENTATION

Most generative tasks do not have the demands for label generation. However, in the stock forecasting task, a clean and informative supervised signal is essential for training the forecaster. According our experiments in Appendix E.3, we suppose that direct generated label fails to serve as the accurate supervised signal for the generated feature. As an alternative, we pave the way to control the synthesis process through guidance inputs, including labels and industry information (Rombach et al., 2022). We can expect that the generated factors will align with the sectors and labels of the original factors, thereby enabling DiffsFormer to generate labels. Our inspiration is drawn from classifier-free guidance (Ho & Salimans, 2022), and since our labels are continuous rather than discrete, we refer to this mechanism as *"predictor-free guidance."*

Technically, according to (Ho & Salimans, 2022), the guiding effect can be achieved by jointly training conditional and unconditional DMs. Specifically, the inference process is in the form of:

$$\hat{\epsilon}_\theta(\mathbf{x}_t, c) = \epsilon_\theta(\mathbf{x}_t, \emptyset) + \omega \cdot (\epsilon_\theta(\mathbf{x}_t, c) - \epsilon_\theta(\mathbf{x}_t, \emptyset)), \tag{10}$$

where $c$ denotes the condition vectors and $\emptyset$ denotes a learnable null vector. During training, $c$ is randomly replaced with $\emptyset$ with a fixed probability to train an unconditional DM. As the guidance strength $\omega$ gets larger, DM receives more rewards when generating $\mathbf{x}_t$ having a high probability $p_\theta(c|\mathbf{x}_t)$. Note that $\omega$ shall be greater than 1 to be effective. The advantages of predictor-free guidance are: 1) it is a simple approach since no auxiliary predictor is needed; 2) it is flexible since it supports other types of conditionings beyond return-ratio labels. In our work, we further explore the use of industry information. We observe that stocks in different industries tend to perform in different patterns. For instance, financial stocks (*e.g.,* banks) usually have low yields but enjoy low volatility, while many information technology stocks have high yields but undertake high volatility. Furthermore, we can synthesize industry-specific data to improve model performance in specific industry sector. One of the unappealing properties of the predictor-free guidance is that it injects the conditionings during the training of DMs. As a result, when adding or modifying conditionings, we need to retrain the DMs although it is time-consuming.

### 3.4 LOSS-GUIDED NOISE ADDITION

We identify that there are certain easy-fitted points within the dataset, and we hypothesize that alleviating the overfitting issues associated with these extreme data points can reduce volatility. § 4.2 illustrates the training loss over time. Notably, the loss for stock forecasting remains quite low during the stock market crash from June 2015 to June 2016, which we suspect is due to the increased proportion of retail investment, characterized by simpler action patterns. A model that overly fits the data from around 2015 is likely to struggle in the present, as market dynamics have become more complex. However, discarding this data is sub-optimal, as it would exacerbate data scarcity.

To address this, we propose a novel strategy termed loss-guided noise addition. Specifically, we utilize training loss as a proxy to introduce stronger noise to data points with lower training loss. As demonstrated in Figure 7c, loss-guided diffusion results in flatter training losses compared to uniform noise addition, effectively alleviating overfitting and decreasing volatility.

### 3.5 MODEL OVERVIEW

Figure 5 elucidates the overall framework of our stock price forecasting model. The framework is designed with several considerations: 1) DMs acts as a plug-and-play data augmentation module, so it can be deployed to different backbones without retraining; 2) our data is organized in sequences, so we explore the use of transformers to better capture the autocorrelation in the sequence, as opposed to the commonly adopted UNet (Ronneberger et al., 2015) in text-to-image generation models; 3) the transfer-based editing framework distills new knowledge while preventing the new data copy from deviating from the original data too much.

## 4 EXPERIMENT

In this section, we conduct experiments on the real-world stock data from 2008 to 2022 provided by (Yang et al., 2020b). Datasets, implementation details, evaluation metrics and trading strategys are shown in Appendix C.

### 4.1 PERFORMANCE COMPARISON

To begin with, we perform a completed comparison between the original and the augmented feature on the mentioned baselines, wherein the percentage of relative improvement on each metric is shown in Table 1 and 2. Note that HIST requires the concept of stocks to build the graph, therefore we don't run it on CSI800 where the concepts are not available. Another notion is that the test time range is 2017-01-01 to 2020-12-31 in previous works (Xu et al., 2021; Wang et al., 2022), which is not consistent with **2020-04-01** to **2022-09-30** in our work. The reason is that we find factors and model performance can decays with age, and we aim to provide with an up-to-date performance of the models. As a result, the performance of backbones in our paper and that in previous works are not comparable. The main observation are as follows:

- In general, the proposed framework DiffsFormer improve the performance of backbone models on average by $0.50\% \sim 13.19\%$ and $4.01\% \sim 70.84\%$ on CSI 300 Index and CSI 800 Index, respectively. Furthermore, our observation aligns with (Zhang et al., 2017a; Taniguchi & Tresp, 1997) that low Signal-to-Noise ratio leads to high variance, for which we conduct significance test. We observe that most of the improvements are significant, while few of them are less significant or even not significant. However, our model has better average performance and lower standard variance which we believe enough to demonstrate the effectiveness of the method.

- For real-world practical use, we could choose the best model on a small validation dataset. We conduct a small experiment: remain train dataset as the $2008\sim2020.04$, and adopt $2020.04\sim2020.12$ to serve as validation dataset and test on $2020.12\sim2022.09$. The test result are shown in columns *Best Ori.* and *Best Ours*, where we observe a remarkable improvement on most of the methods.

- Excess Return is the primary performance metric since the ultimate goal of stock forecasting is to achieve substantial profits. Besides, we also adopt Weighted-IC (§2) to better characterize the correlation between the prediction and label on top stocks. From the table, we can observe that: 1) Weighted-IC for CSI800 is obviously lower than that for CSI300, which is consistent with excess return performance in Table 1 and 2. 2) The models' rankings in terms of weighted-IC and excess return are similar, especially on CSI800, suggesting weighted IC can be served as a metric to measure the potential of reaching a high excess return. 3) DiffsFormer boosts the Weighted-IC for most of the methods on the CSI300 and improves the Weighted-IC for more than half of the methods on the CSI800, verifying its effectiveness of improving model performance. We also report IC and RankIC in Appendix E, but we don't think it is always positively associated to the excess return. Accurate prediction of high-volatility (top and bottom) stocks are more important to acquire profits and avoid losses, as shown in Figure 6. Our model has a lower MSE and RMSE

Table 1: Excess return and Weighted-IC comparison on **CSI300**. The better results are indicated in boldface. Deep blue boxes indicates passing 0.05 level test. Shallow blue boxes indicates passing 0.2 level test. Shallow yellow boxes indicates failing significance test.

| | Excess Return | | | | | | Weighted-IC | | |
|---|---|---|---|---|---|---|---|---|---|
| | Original | Ours | *Improv.* | p-value | *Best Ori.* | *Best Ours* | Original | Ours | *Improv.* |
| **MLP** | $0.2093_{\pm 0.0300}$ | $\mathbf{0.2163}_{\pm 0.0210}$ | 3.34% | 0.123 | 0.2278 | **0.2345** | $0.0326_{\pm 0.0023}$ | $\mathbf{0.0332}_{\pm 0.0021}$ | 1.84% |
| **LSTM** | $0.2312_{\pm 0.0308}$ | $\mathbf{0.2336}_{\pm 0.0219}$ | 1.04% | 0.868 | 0.2498 | **0.2587** | $0.0295_{\pm 0.0032}$ | $\mathbf{0.0339}_{\pm 0.0025}$ | 14.92% |
| **GRU** | $0.2161_{\pm 0.0293}$ | $\mathbf{0.2413}_{\pm 0.0149}$ | 11.66% | 0.157 | **0.2167** | 0.2140 | $0.0324_{\pm 0.0012}$ | $\mathbf{0.0383}_{\pm 0.0011}$ | 18.21% |
| **SFM** | $0.2189_{\pm 0.0325}$ | $\mathbf{0.2200}_{\pm 0.0175}$ | 0.50% | 0.923 | 0.2253 | **0.2289** | $0.0288_{\pm 0.0029}$ | $\mathbf{0.0300}_{\pm 0.0030}$ | 4.17% |
| **GAT** | $0.2461_{\pm 0.0176}$ | $\mathbf{0.2701}_{\pm 0.0168}$ | 9.75% | 0.019 | 0.2333 | **0.3021** | $\mathbf{0.0354}_{\pm 0.0006}$ | 0.0324 $_{\pm 0.0004}$ | -8.47% |
| **ALSTM** | $0.2047_{\pm 0.0351}$ | $\mathbf{0.2317}_{\pm 0.0233}$ | 13.19% | 0.012 | 0.2410 | **0.2757** | $0.0260_{\pm 0.0038}$ | $\mathbf{0.0312}_{\pm 0.0033}$ | 20.00% |
| **HIST** | $0.2272_{\pm 0.0352}$ | $\mathbf{0.2410}_{\pm 0.0207}$ | 6.07% | 0.249 | **0.2420** | 0.2243 | $0.0249_{\pm 0.0066}$ | $\mathbf{0.0317}_{\pm 0.0026}$ | 27.31% |
| **MTMD** | $0.2129_{\pm 0.0355}$ | $\mathbf{0.2547}_{\pm 0.0207}$ | 19.63% | 0.024 | 0.1408 | **0.1830** | $0.0316_{\pm 0.0027}$ | $\mathbf{0.0347}_{\pm 0.0021}$ | 27.31% |
| **Transformer** | $0.2789_{\pm 0.0376}$ | $\mathbf{0.3127}_{\pm 0.0113}$ | 12.12% | 0.016 | 0.2688 | **0.3360** | $0.0387_{\pm 0.0038}$ | $\mathbf{0.0433}_{\pm 0.0048}$ | 11.89% |

Table 2: Performance comparison on **CSI800**. The better results are indicated in boldface.

| | Excess Return | | | | | | Weighted-IC | | |
|---|---|---|---|---|---|---|---|---|---|
| | Original | Ours | *Improv.* | p-value | *Best Ori.* | *Best Ours* | Original | Ours | *Improv.* |
| **MLP** | $0.1037_{\pm 0.0383}$ | $\mathbf{0.1161}_{\pm 0.0223}$ | 11.96% | 0.102 | **0.1292** | 0.1243 | $0.0052_{\pm 0.0041}$ | $\mathbf{0.0063}_{\pm 0.0032}$ | 21.15% |
| **LSTM** | $0.1248_{\pm 0.0282}$ | $\mathbf{0.1298}_{\pm 0.0317}$ | 4.01% | 0.758 | 0.1165 | **0.1408** | $\mathbf{0.0075}_{\pm 0.0055}$ | 0.0024 $_{\pm 0.0026}$ | -68.00% |
| **GRU** | $0.0758_{\pm 0.0307}$ | $\mathbf{0.1295}_{\pm 0.0292}$ | 70.84% | 3e-4 | 0.0828 | **0.1265** | $0.0005_{\pm 0.0027}$ | $\mathbf{0.0128}_{\pm 0.0029}$ | 2460.00% |
| **SFM** | $0.0906_{\pm 0.0413}$ | $\mathbf{0.1250}_{\pm 0.0375}$ | 37.97% | 0.004 | 0.0980 | **0.1415** | $\mathbf{0.0028}_{\pm 0.0032}$ | 0.0026 $_{\pm 0.0030}$ | -7.14% |
| **GAT** | $0.1814_{\pm 0.0309}$ | $\mathbf{0.2013}_{\pm 0.0210}$ | 10.97% | 0.007 | 0.0849 | **0.0862** | $\mathbf{0.0083}_{\pm 0.0010}$ | 0.0047 $_{\pm 0.0008}$ | -43.37% |
| **ALSTM** | $0.1030_{\pm 0.0253}$ | $\mathbf{0.1518}_{\pm 0.0290}$ | 50.29% | 5e-4 | 0.0880 | **0.2257** | $0.0025_{\pm 0.0064}$ | $\mathbf{0.0094}_{\pm 0.0023}$ | 276.00% |
| **Transformer** | $0.1751_{\pm 0.0386}$ | $\mathbf{0.1903}_{\pm 0.0382}$ | 8.68% | 0.280 | 0.1583 | **0.2923** | $0.0066_{\pm 0.0058}$ | $\mathbf{0.0159}_{\pm 0.0054}$ | 140.91% |

in high-volatility stocks, although got worse overall metrics. The reason is that our target domain (CSI300 and CSI800) consists of more established companies with stable earnings, and tend to have lower volatility; the source domain consists of all stocks in China A-share, which means the source domain have a higher volatility than the target one. Knowledge distillation enhances the prediction ability of high-volatility stocks at the expense of the low-volatility ones. Since our strategy is discovering Top-30 stocks, this property is promising and leads to higher profit.

## 4.2 EFFECTIVENESS ANALYSIS

In this section, we will discuss each component of DiffsFormer, including loss-guided diffusion, transfer diffusion, conditional diffusion, and comparison with other augmentation algorithms.

**Editing Mechanism.** As the financial data is noisy, recall that we restrict the perturb step to a small value $T' \ll T$, where $T$ is the diffusion step and $T'$ is the editing step. $T'$ could control the strength of knowledge distillation: a larger $T'$ makes generated data resemble more feature distribution from the source domain, while a smaller $T'$ makes edited data closer to original target domain data. To support this argument, we report the editing steps along with corresponding model performance and FID between the original and the edited data in Table 3. We observe a trade-off between model performance and the editing step, which we attribute to the increased data diversity in the very early diffusion steps and the decreased data fidelity in the later steps. We also conduct experiment comparison on direct generation, random noise addition and editing, observing that generated data are restricted to locate near the original data when we edit the existing sample from the target domain. The detailed experiment can be found in Appendix E.1.

Table 3: The Effect of Editing Steps

| Steps | 200 | 300 | 400 | 500 |
|---|---|---|---|---|
| Performance | 0.2843 | 0.3127 | 0.2936 | 0.2712 |
| W-Distance | 0.4113 | 0.6908 | 1.1380 | 1.8927 |

**Loss-guided diffusion.** Besides the excess return, information ratio (IR)[1] is another essential measure of the stock forecasting performance which measures the stability and generalization of the model. In Figure 7d, we observe that (1) data augmentation can increase the IR of the model; (2)

---

[1]https://en.wikipedia.org/wiki/Information_ratio

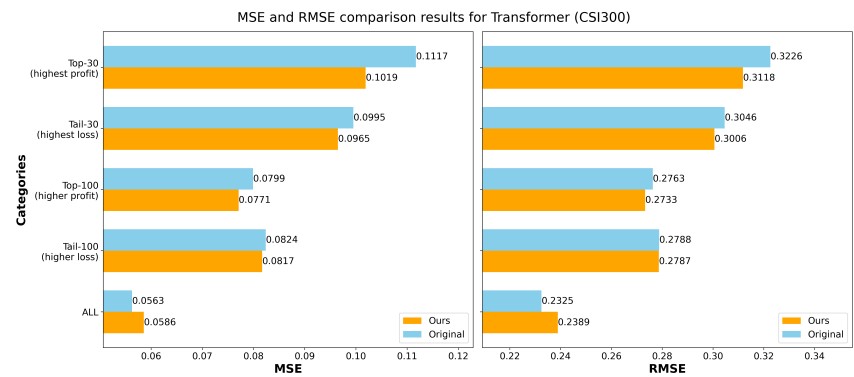

Figure 6: MSE and RMSE comparision. Top/Tail-30(100) category represents the 30(100) stocks exhibited the highest price increases/decreases.

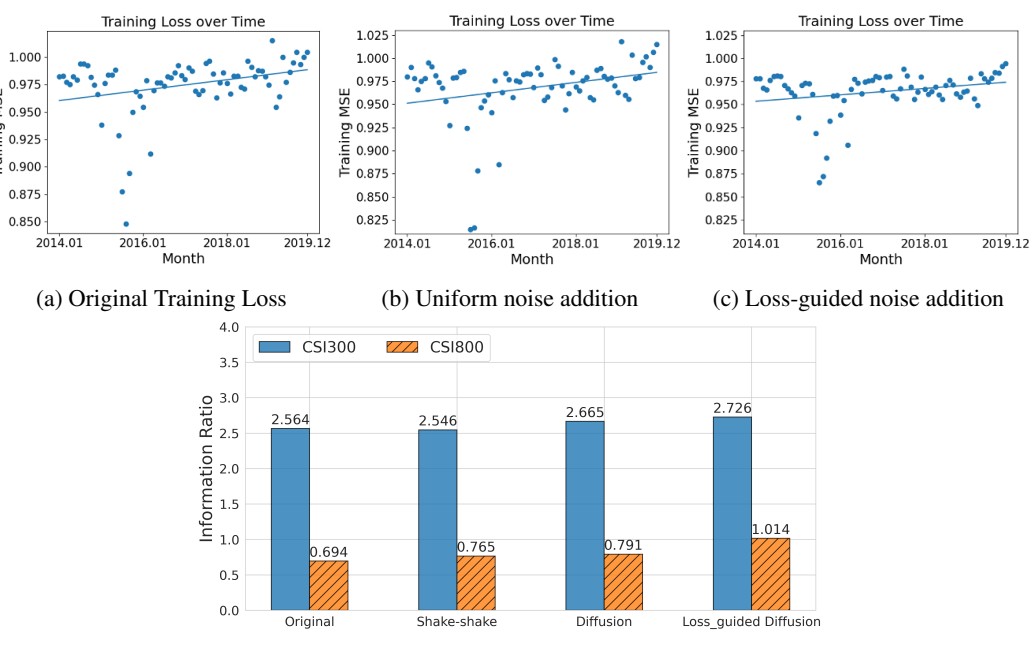

(a) Original Training Loss    (b) Uniform noise addition    (c) Loss-guided noise addition

(d) Information ratio by different training data.

Figure 7: The illustration of the impact of loss-guided diffusion.

DM outperforms shake-shake (Gastaldi, 2017), another data augmentation method based on Transformer; (3) loss-guided diffusion can further increase IR and decrease the volatility.

**Transfer Diffusion.** Recall that in § 3.1, we design a novel inference process to distill new knowledge to generated data through transfer learning. To verify the real cause of performance improvement, we aim to exclude the interference of the new information. The result is shown in Table 4, where the fine tuning column denotes the mechanism of training on source domain and testing on target domain, and diffusion DA stands for diffusion-based data augmentation with transfer learning. Observations are threefold: 1) While training in a larger source domain before fine-tuning in the target domain introduces new informa-

Table 4: Diffusion-based Data augmentation and Fine Tuning results. CSIS denotes all stocks in China A-Share.

| Target Domain | Source Domain | Fine Tuning | Diffusion DA |
|---|---|---|---|
| CSI800 | CSI800 | $0.1751_{\pm 0.0386}$ | $0.1793_{\pm 0.0113}$ |
| | CSIS | $0.1641_{\pm 0.0300}$ | $0.1903_{\pm 0.0382}$ |
| CSI300 | CSI300 | $0.2789_{\pm 0.0376}$ | $0.2861_{\pm 0.0547}$ |
| | CSI800 | $0.2773_{\pm 0.0181}$ | $0.2789_{\pm 0.0333}$ |
| | CSIS | $0.2432_{\pm 0.0372}$ | $\mathbf{0.3127}_{\pm 0.0113}$ |

tion, it may degrade model performance due to differences in distribution between the two

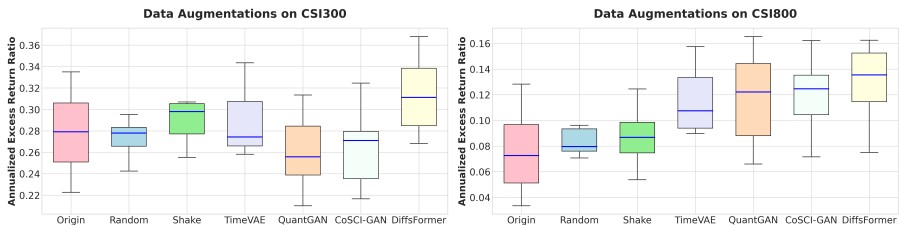

Figure 8: Comparison between different augmentation methods with Transformer and GRU.

domains. 2) When the source and target domains are identical, meaning no new information is introduced, DM still enhances performance. 3) Transfer diffusion significantly boosts model performance, underscoring the effectiveness of the transfer learning mechanism.

**Conditional Diffusion.** Conditionings are incorporated in DiffsFormer for two reasons: (1) Help generate the corresponding label; (2) Boost the performance. The stock forecasting performance with different conditionings are reported in Table 5, where PFG stands for predictor-free guidance and ER stands for Excess Ratio. We observe that DMs achieve lower Wasserstein distance and contribute to a better model performance with the help of conditionings including ER and sector. Additionally, **fidelity and diversity trade-off *w.r.t.* guidance strength** are shown in Appendix E.2, consistent with previous works, we observe data fidelity increases and data diversity decreases when the guidance strength increases.

Table 5: Performance with Different Conditionings.

|  |  | Performance | Wasserstein |
|---|---|---|---|
| w/o Diffusion |  | 0.2789 | - |
| No Conditioning |  | 0.2919 | 0.9009 |
| PFG | ER | 0.2971 | 0.7335 |
|  | Sector | 0.3009 | 0.8226 |
|  | ER + Sector | **0.3127** | **0.6908** |

**Comparison with Other Augmentation Algorithms.** In this work, we reveal that data augmentation plays a pivotal role in stock forecasting. And in this section, we aim to verify the DiffsFormer's superiority over other data augmentation mechanisms. The experimental results are reported in Figure 8. The baselines for the methods are listed in Appendix C.2. From Figure 8, we observe that: 1) Time-series generation methods like Quant-Gan, TimeVAE, COSCI-GAN fails to improve the performance of vanilla model. We suppose the reasons are two fold: these models do not generate labels and do not have conditionings hence they fall short in feature-label matching; these models are mostly univariate methods (Kollovieh et al., 2023), thus they overlook the correlations between multivariate variables in our task. 2) Shake-shake and DiffsFormer are two effective data augmentation mechanisms that outperform the random gaussian noise addition, and our proposed method DiffsFormer performs better than Shake-shake by a large margin. 3) Data augmentation can enhance the model stability, as the box of the augmentation is commonly shorter than that of the original. 4) Diffsformer has the best worst-case model performance.

## 5 CONCLUSION AND LIMITATIONS

**Conclusion.** We address the critical challenge of data scarcity in stock forecasting by introducing DiffsFormer. Our approach augments stock factors using label and sector information, while incorporating transfer learning in a Diffusion Model framework. By training on a larger source domain and synthesizing with target domain data, DiffsFormer effectively distills new knowledge, mitigating data limitations and enhancing forecasting accuracy. This work pioneers data augmentation in stock forecasting using diffusion models, opening avenues for future research. We find that conditioning on factors like industry sectors enhances performance, suggesting potential for targeted improvements through factor editing or generating stocks with specific attributes. Our study also underscores the challenges of homogeneity in stock forecasting. Limitations are listed in E.4.

## 6 ETHICS STATEMENT

Acknowledging the potential impact on stakeholders, we advocate for responsible investment practices and compliance with privacy laws. We are committed to continuous improvement and welcome feedback to address any emerging ethical concerns in our work.

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

## A  ALGORITHMS

---

**Algorithm 1: DiffsFormer Training**

---

    **Input:** stock data $\boldsymbol{X} \in \mathbb{R}^{n \times d \times k}$, diffusion step $T$
    **for** $t = 1$ to $T$ **do**
        initialize $\beta_t$ and calculate $\overline{\alpha}_t$
    **end for**
    Select an editing step $T' \leq T$
    **while** Not Converge **do**
        $i \sim \text{Uniform}\{1, 2, \cdots, \text{n}\}$
        $t \sim \text{Uniform}\{1, 2, \cdots, \text{T}'\}$
        $\epsilon \sim \mathcal{N}(0, mI)$
        $\mathbf{x}_0 := \boldsymbol{X}[i]$
        calculate $\mathbf{x}_t$ given $\mathbf{x}_0$ with equation 6
        calculate $\mathcal{L}_{da}$ with equation 8
        Take a gradient step on $\nabla_\theta \mathcal{L}_{da}$
    **end while**

---

**Algorithm 2: DiffsFormer Inference**

---

    **Input:** number of data to be generated $m$, sampling steps $l$, conditionings $c$ (if guidance is enabled), editing step $T'$ selected during training
    **while** Generated point $< m$ **do**
        $i \sim \text{Uniform}\{1, 2, \cdots, \text{n}\}$
        $\mathbf{x}_0 := \boldsymbol{X}[i]$
        calculate $\mathbf{x}_{T'}$ given $\mathbf{x}_0$ with equation 6
        $\mathbf{x}_{\tau_l} := \mathbf{x}_{T'}$
        **for** $t = l$ to $0$ **do**
            calculate $\mathbf{x}_{\tau_{t-1}}$ with DDIM sampling
        **end for**
    **end while**

---

## B  DENOISING DIFFUSION PROBABILISTIC MODEL

Denoising Diffusion Probabilistic Models (DDPM) have achieved impressive performance in various domains, especially in text-to-image scenarios (Nichol et al., 2021; Ramesh et al., 2022). Typically, training a DM needs diffusion and denoising processes.

**Diffusion process.** Given a data point $\mathbf{x}_0 \sim q(\mathbf{x}_0)$, the diffusion process gradually adds noise to construct a sequence of step-dependent variables $\{\mathbf{x}_t\}_{t=1}^T$ (Wang et al., 2023) which forms a Markov chain as (Tashiro et al., 2021):

$$q(\mathbf{x}_{1:T}|\mathbf{x}_0) = \prod_{t=1}^T q(\mathbf{x}_t|\mathbf{x}_{t-1}), \tag{11}$$

where $q(\mathbf{x}_t|\mathbf{x}_{t-1}) = \mathcal{N}(\mathbf{x}_t; \sqrt{\alpha_t}\mathbf{x}_{t-1}, \beta_t\boldsymbol{I})$. $\mathcal{N}$ denotes the Gaussian distribution, $\alpha_t$ controls the strength of signal retention, and $\beta_t$ controls the scale of the added noise. These two scalars are predefined for each step $t$, and one commonly used setting is the variance preserving process (Ho et al., 2020) where $\alpha_t = 1 - \beta_t$.

**Denoising process.** The goal of the denoising process is to reconstruct the corresponding noise vector by inverting the transformations performed in the diffusion process. This process is defined by another Markov chain (Tashiro et al., 2021):

$$p_\theta(\mathbf{x}_{0:T}) = p(\mathbf{x}_T) \prod_{t=1}^T p_\theta(\mathbf{x}_{t-1}|\mathbf{x}_t), \tag{12}$$

where $\mathbf{x}_T \sim \mathcal{N}(0, \boldsymbol{I})$. $p_\theta$ is the distribution estimation of $q$, for which $p_\theta(\mathbf{x}_{t-1}|\mathbf{x}_t) = \mathcal{N}(\mathbf{x}_{t-1}; \mu_\theta(\mathbf{x}_t, t), \sigma_\theta(\mathbf{x}_t, t)\boldsymbol{I})$. Concretely, for each sample in the batch, a time step $t$ is uniformly sampled from $\{1, 2, ..., T\}$, followed by the adjustment of the noise at time $t$.

**Inference process.** Once $\theta$ is well-trained, the DM can generate samples from the standard Gaussian distribution with $\mathbf{x}_T \sim \mathcal{N}(0, \boldsymbol{I})$ and then repeatedly recover $\mathbf{x}_T \to \cdots \to \mathbf{x}_t \to \mathbf{x}_{t-1} \to \cdots \to \mathbf{x}_0$ given $p_\theta(\mathbf{x}_{t-1}|\mathbf{x}_t)$. As $T \to \infty$, the generative process modeled with Gaussian conditional distributions becomes a good approximation.

## C    REPRODUCIBILITY

In this subsection, we introduce some details of the proposed work for easier reproduction.

### C.1    REPRODUCIBILITY STATEMENT

All the results in this work are reproducible. We'll provide codes for the DiffsFormer upon acceptance. In following sections, we will discuss hyperparameters search space, optimal hyperparameters, details about data preprocessing, and software/hardware.

### C.2    BASELINES

To verify the performance of the proposed framework in stock forecasting, we employ eight commonly used machine learning models as forecasting backbones:

- **LSTM** (Hochreiter & Schmidhuber, 1997): a Long Short-Term Memory network based stock price forecasting method.
- **GRU** (Chung et al., 2014): a Gated Recurrent Unit (GRU) network based stock price forecasting method.
- **SFM** (Zhang et al., 2017b): a State Frequency Memory (SFM) network that decomposes the hidden states of memory cells into multiple frequency components to model different latent trading patterns.
- **GAT** (Velickovic et al., 2018): Graph attention network (GAT) is utilized to aggregate the stock node embeddings attentively.
- **ALSTM** (Feng et al., 2019): an LSTM variant that incorporates temporal attentive aggregation layer to aggregate information from hidden embeddings in previous timestamps.
- **Transformer** (Vaswani et al., 2017): transformer-based stock forecasting model.
- **HIST** (Xu et al., 2021): a graph-based framework that mines the concept-oriented shared information from predefined concepts and hidden concepts.

The baselines in Figure 8 are:

- **Shake-shake** (Gastaldi, 2017): a stochastic affine combination of the multi-branch network
- **TimeVAE** (Desai et al., 2021): a novel architecture for synthetically generating time-series data with the use of Variational AutoEncoders (VAEs).
- **QuantGAN** (Wiese et al., 2019): generative adversarial networks (GAN) that utilizes temporal convolutional networks (TCNs) to capture time-series dependencies.
- **COSCI-GAN** (Seyfi et al., 2022): a novel GAN framework that takes time series' common origin into account and favors channel/feature relationships preservation.

### C.3    DATASET

Following (Xu et al., 2021; Wang et al., 2022), we evaluate the proposed framework on two real-world stock datasets: CSI 300 and CSI 800. The CSI 300 comprise the largest 300 stocks traded on the Shanghai Stock Exchange and the Shenzhen Stock Exchange[2], and represents the performance of the whole A-share market in China. CSI 800 is a larger dataset consisting of CSI 500 and CSI 300, aiming to add some stocks with smaller size. Note that DiffsFormer aims at editing the existing samples with new information from a larger domain. Hence in practice, we use all stocks in the China A-share market to train the DM and editing on CSI 300 and CSI 800, respectively.

---

[2]https://en.wikipedia.org/wiki/CSI_300_Index

## C.4 Evaluation Metrics

**Annualized Excess Return** is served as the primary evaluation metric. Besides, **Weighted IC** is adopted to reflect the predictive power of the models. To eliminate performance fluctuation, we run the training and testing procedure 8 times for all of the methods and report the average value and the standard deviation. Since the training of DMs and the predictor is decoupled, we only run DM once for time efficiency. Furthermore, we also adopt Mean Squared Error (MSE) and Root Mean Squared Error (RMSE) to indicate the predict ability of the models. **Weighted IC** is adopted to reflect the predictive power of the models. To eliminate performance fluctuation, we run the training and testing procedure 8 times for all of the methods and report the average value and the standard deviation. Since the training of DMs and the predictor is decoupled, we only run DM once for time efficiency.

## C.5 Trading Strategy

Our stock trade adopts "top30drop30" strategy: "top30" means that we keep the stocks with top30 predicted scores; and "drop30" means that each stock will be droped if its score falls out of top30, regardless of its previous performance.

## C.6 Factors

We use the Alpha158 factors provided by the AI-oriented quantitative investment platform Qlib[3]. These factors review the basic stock information including *kbar, price, volume, and some rolling factors* in different time windows. For each stock at date $t$, we look back 8 days to construct a sequence of factor as $\mathbf{x} \in \mathbb{R}^{8 \times 158}$. During the time span between 2008-01-01 and 2022-09-30, the number of sequences is 2109804. Hence our input matrix $X$ is of shape $2109804 \times 8 \times 158$.

## C.7 Model Parameters

We carefully search the hyper-parameters over the search range. The optimal parameters are reported in Table 6.

Table 6: Hyper-parameters and the search range, the optimal parameters are indicated in boldface.

| Parameters | Search Range |
|---|---|
| editing step during inference | {200, **300**, 400, 500} |
| layers in DM | {3, **6**} |
| stop loss thred | {0.6, 0.8, 0.9, 0.95, **0.965**, 1} |
| batch norm | {**False**, True} |
| norm first | {False, **True**} |
| guidance strength | {1.1, 2, **3**, 4} |
| sector condition | {False, **True**} |
| label condition | {False, **True**} |

## C.8 Data Preprocessing

**Robust Z-score Normalization.** Generally, the values between factors are not in the same scale. To address this issue, we adopt *Robust Z-score Normalization* within stocks. Based on z-score, robust z-score replace mean and standard deviation with median (MED) and the median absolute deviation (MAD). In robust statistical methods [4], MED is the robust measure of central tendency, while mean is not; MAD is robust measure of statistical dispersion, while standard deviation is not. Specifically, the $i$th input stock data is normalized to:

$$\hat{\mathbf{x}}[i] = |\mathbf{x}[i] - \mathrm{MED}(\mathrm{X})| / \mathrm{MAD}(\mathrm{X}).  \tag{13}$$

---

[3]https://github.com/microsoft/qlib
[4]https://en.wikipedia.org/wiki/Robust_statistics

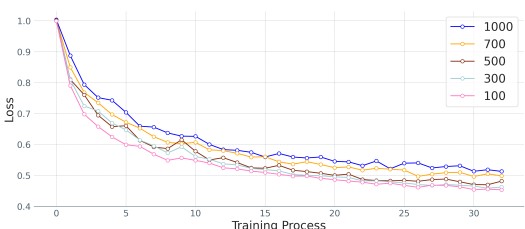

Figure 9: Loss curves with different sampling steps.

**Dropping Extreme Label.** When a stock hits limit up(limit down), stockholders are more reluctant to sell (buy) at this time, for they expect that the trend continues; as a result, it is difficult for other stockholders to buy (sell). Therefore, it is meaningless for the model to learn to "buy when there is a limit up, and sell when there is a limit down". To tackle this challenge, we propose to drop the extreme label to exclude the influence of extreme values. It is achieved in two ways: 1) we set a upper threshold and a lower threshold; 2) we drop the first and the last few percent labels.

## C.9 SOFTWARE AND HARDWARE

DiffsFormer is implemented with Python 3.8.16, Pytorch 1.11.0. All of the backbones are implemented with the open-sourced code in Qlib. We run the experiments on servers equipped with NVIDIA Tesla V100 GPU and 2.50GHz Intel Xeon Platinum 8163 CPU.

## C.10 MORE ARCHITECTURE IMPROVEMENTS

Following (Peebles & Xie, 2022), we adjust the transformer structure to fit predictor-free guidance. Specifically, as shown in Figure 5, we feed the diffusion steps into the transformer module. Furthermore, it contains an adaptive layer norm module and a zero-initialized scalar module.

**Time step encoding.** During the training, DiffsFormer needs to know which diffusion step it is trained for. In our work, we encode the current diffusion step into *Sinusoidal positional encoding* and add it to the feed forward network. This embedding scheme is appropriate to encode the time step information, the positional encoding at position $p + k$ can be linearly represented by the positional encoding at position $p$.

**Adaptive layer norm (ALN).** Adaptive layer norm (Perez et al., 2018) are widely adopted in generative models (Dhariwal & Nichol, 2021; Brock et al., 2019). In the transformer block, we append an additional ALN layer which regresses the scale and shift parameters $\gamma$ and $\beta$ from two affine transformations $f$ and $h$ to the sum of conditioning vectors.

**Zero initialization.** In addition to ALN layer, many DMs (Dhariwal & Nichol, 2021; Ho et al., 2020; Peebles & Xie, 2022) also incorporate zero initialization in the framework, meaning that the model parameters are initialized as zero such that the conditioning is ineffective when training just starts. In this case, MLP is initialized to make $\alpha_1$ and $\alpha_2$ equal to 0, and thus transformer block becomes an identity function (Peebles & Xie, 2022; Goyal et al., 2017) (*i.e.,* input tokens are directly fed to the next layer).

**Time improvement.** As stated in §3.2, we initialize $\alpha$ and $\beta$ with total diffusion steps $T$ to ensure correctness; however, we sample training step $t$ from $Uniform\{1, 2, \cdots, T'\}$ instead of $Uniform\{1, 2, \cdots, T\}$: compared to traditional DM, the probability of sampling useful steps that are smaller than $T'$ is increased. The loss curves with maximum sampling steps within the set $\{100, 300, 500, 700, 1000\}$ are elucidated in Figure 9. Note that the figure represents the average loss within sampling step 100 instead of diffusion step 1000. We discover that with the decrease of sampling steps, DMs embrace with a more sharp loss curve, which means they can converge faster.

## D RELATED WORKS

In this section, we will introduce related works in stock forecasting.

Stock forecasting is a field that utilizes historical time-series data to predict future stock prices. Machine learning models, particularly time-series models such as LSTM, GRU, and Bi-LSTM, have gained popularity in this domain (Zou et al., 2022; Hochreiter & Schmidhuber, 1997; Chung et al., 2014; Graves & Schmidhuber, 2005; Xia et al., 2024).

Researchers have proposed tailored models to better fit the financial scenario. For example, Li et al. (Li et al., 2018) introduce extra input gates to extract positive and negative correlations between factors. Ding et al. (Ding & Qin, 2020) propose a novel LSTM model to simultaneously predict the opening, lowest, and highest prices of a stock. Agarwal et al. (Rather et al., 2015) propose a hybrid prediction model (HPM) that combines three time-series models. Zhang et al. (Zhang et al., 2017b) propose a State Frequency Memory (SFM) network that decomposes the hidden states of memory cells into multiple frequency components to model different latent trading patterns. Feng et al. (Feng et al., 2019) incorporate a temporal attentive aggregation layer and adversarial training into an LSTM variant. Chen et al. (Chen et al., 2019) use Bi-LSTM to encode stock data and financial news representations in their SSPM and MSSPM models.

CNNs are also believed to capture important features for predicting stock fluctuations. For instance, Deng et al. (Deng et al., 2019) propose the Knowledge-Driven Temporal Convolutional Network (KDTCN), which integrates knowledge graphs with CNNs to fully utilize industrial relations. Lu et al. (Lu et al., 2021) enhance a CNN-based model by extracting historical influential stock fluctuations with attention mechanism. Chandar (Chandar, 2022) transforms technical indicators into images and used them as input for a CNN model.

To handle non-Euclidean structured data, some researchers have incorporated Graph Neural Networks (GNNs) into stock forecasting. Velickovic et al. (Velickovic et al., 2018) construct a graph with stocks as nodes and used graph attention network (GAT) to aggregate neighbor embeddings. Xu et al. (Xu et al., 2022) construct a stock market relationship graph and extracted information hierarchically. Li et al. (Li et al., 2020) propose an LSTM Relational Graph Convolutional Network (LSTM-RGCN) model that handles both positive and negative correlations among stocks.

The Transformer model (Vaswani et al., 2017), with self-attention and positional encoding mechanisms, has shown great potential in stock forecasting. Ding et al. (Ding et al., 2020) improve the Transformer by incorporating multi-scale Gaussian prior, optimizing locality, and implementing Orthogonal Regularization. Yoo et al. (Yoo et al., 2021) propose a Data-axis Transformer with Multi-Level Contexts (DTML) to learn the correlations between stocks. Yang et al. (Yang et al., 2020a) introduce the Hierarchical, Transformer-based, multi-task (HTML) model for predicting short-term and long-term asset volatility. FTS-Diffusion (Huang et al., 2024) consists of three modules to model irregular and scale-invariant patterns and generate synthetic financial time series.

# E  MORE RESULTS

## E.1  EDITING V.S. GENERATING

In Figure 10, following recent work (Shipard et al., 2023), we visualize the relationship between the augmented features and the original stock features in blue and pink, respectively. We have two observations: 1) Comparing Figure 10a and Figure 10c, we find generated data are restricted to locate near the original data when we edit the existing sample from the target domain; while many points deviate the target domain distribution when we directly synthesize new data points. 2) Random gaussian noise addition can be treated as a special augmentation mechanism. We run several experiments with different level random gaussian noise addition and plot in Figure 10b the t-SNE of feature distribution with the most accurate return ratio prediction. Our proposed method looks better than random noise addition.

## E.2  DATA FIDELITY AND DIVERSITY

Our work adopts diffusion-based data augmentation module to synthesize data, which helps alleviate the serious data scarcity issue in stock forecasting. Particularly, before the start of training of the predictor in each epoch, we generate a new set of stock data. Therefore, the total amount of data utilized is $n\times$ the original where $n$ is equal to the total number of epoches. In other words, with DiffsFormer, the backbone can observe $n\times$ the data for once, instead of observing original data

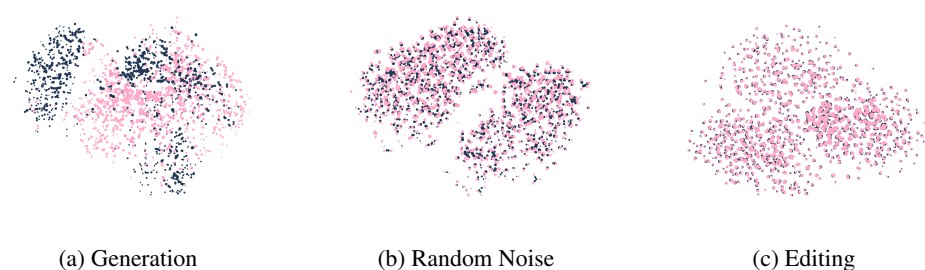

(a) Generation  (b) Random Noise  (c) Editing

Figure 10: t-SNE plots of original features (pink) and augmented features (blue) elucidating the effect of editing.

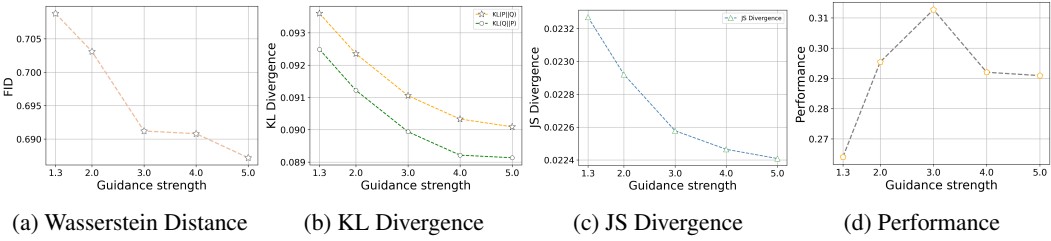

(a) Wasserstein Distance  (b) KL Divergence  (c) JS Divergence  (d) Performance

Figure 11: Illustration of Data Fidelity and Diversity *w.r.t.* Guidance Strength

for $n$ times. To measure the fidelity and diversity of the data generated, two metrics are commonly adopted (Ho et al., 2020; Rombach et al., 2022; Peebles & Xie, 2022): Fréchet Inception Distance (FID) and Inception Score (IS). However, they require a well-trained classifier (*e.g.,* Inception Network). Hence we directly calculate the Wasserstein Distance between generated and original data to serve as an alternate for FID. As reported in Figure 11a, 11b and 11c, the distance between original and generated data decreases as the guidance strength gets stronger, suggesting a high fidelity of the generated data. In addition, from Figure 11d, we find that strong guidance strength may lead to performance drop. We attribute the reason for the phenomenon to the lack of diversity of the data.

### E.3 FEASIBILITY OF LABEL GENERATION

We have conducted experiments to validate the feasibility of directly generating labels. Taking Figure 12 as an example, we calculate the $R^2$ score between the augmented and the original factors (label). It is observed that the label has the least correlation among the 159 dimensions. Furthermore, we report an experimental comparison between the settings of label-generation and label-conditioning in Table 7. Directly appending the label to the factor vector is ineffective.

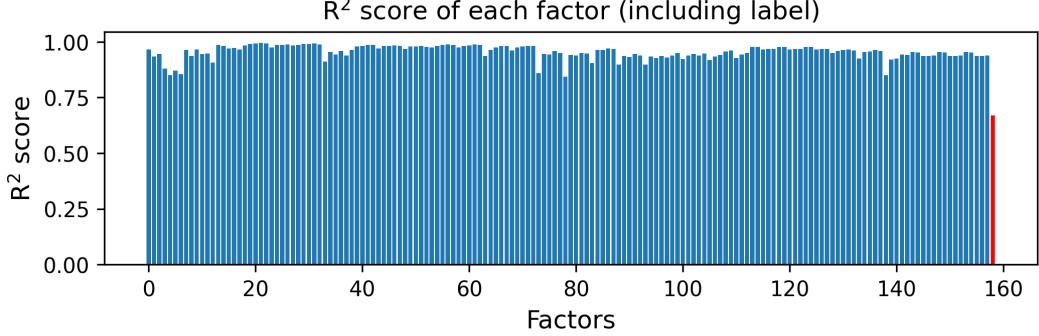

Figure 12: The $R^2$ score between the generated and the original factors and label. $R^2$ score is the square of the Pearson Correlation. The blue bars represent the $R^2$ scores of 158 factors, while the red bar shows the $R^2$ score of the label.

Table 7: Model performance with label-generation and label-condition mechanisms

|  | Label-generation | Label-condition |
|---|---|---|
| Model Performance | 0.159327 | 0.312679 |

### E.4 LIMITATIONS

We believe direction prediction also faces with the scarcity problem as we share the same input data type, hence Diffsformer may help with the direction prediction task from this perspective. However, since we didn't delve deep into this task, we may assume that direction prediction may have a higher demand for feature-label matching. DiffsFormer uses the original label for the generated feature, we suppose it's OK for the regression task, but we're not sure if it works for the classification task. We believe developing ways to generate real label or enhance feature-label matching would be helpful. (2) Portfolio management may involve Deep Reinforcement Learning (DRL) approaches. We think DRL requires expert knowledge in reward design, policy selection and rich experience in optimization to which we lack the capacity. Our strategy now is simple: a money-weighted position over the predicted top-30 stocks. However, we believe a better selection strategy would be helpful to the buy and sell decision.

Table 8: Performance comparison on CSI300. The better results are indicated in boldface.

| Methods | CSI300 | | | | | |
|---|---|---|---|---|---|---|
|  | IC | | | RankIC | | |
|  | Original | Ours | *Improv.* | Original | Ours | *Improv.* |
| MLP | $0.0508_{\pm0.0044}$ | $\mathbf{0.0537_{\pm0.0026}}$ | 5.71% | $0.0499_{\pm0.0059}$ | $\mathbf{0.0509_{\pm0.0034}}$ | 2.00% |
| LSTM | $\mathbf{0.0516_{\pm0.0022}}$ | $0.0429_{\pm0.0026}$ | -16.86% | $\mathbf{0.0519_{\pm0.0021}}$ | $0.0455_{\pm0.0021}$ | -12.33% |
| GRU | $\mathbf{0.0536_{\pm0.0038}}$ | $0.0511_{\pm0.0012}$ | -4.66% | $\mathbf{0.0552_{\pm0.0037}}$ | $0.0516_{\pm0.0012}$ | -6.52% |
| SFM | $0.0505_{\pm0.0018}$ | $\mathbf{0.0510_{\pm0.0025}}$ | 0.99% | $0.0507_{\pm0.0026}$ | $\mathbf{0.0526_{\pm0.0029}}$ | 3.75% |
| GAT | $\mathbf{0.0558_{\pm0.0012}}$ | $0.0532_{\pm0.0007}$ | -4.66% | $0.0540_{\pm0.0014}$ | $\mathbf{0.0551_{\pm0.0006}}$ | 2.04% |
| ALSTM | $\mathbf{0.0502_{\pm0.0027}}$ | $0.0450_{\pm0.0023}$ | -10.36% | $\mathbf{0.0510_{\pm0.0031}}$ | $0.0439_{\pm0.0019}$ | -13.92% |
| HIST | $\mathbf{0.0547_{\pm0.0011}}$ | $0.0518_{\pm0.0032}$ | -5.30% | $\mathbf{0.0545_{\pm0.0023}}$ | $0.0535_{\pm0.0025}$ | -1.83% |
| MTMD | $\mathbf{0.0495_{0.0024}}$ | $0.0476_{\pm0.0023}$ | -3.84% | $\mathbf{0.0488_{\pm0.0040}}$ | $0.0466_{\pm0.0031}$ | -4.51% |
| Transformer | $0.0598_{\pm0.0031}$ | $\mathbf{0.0603_{\pm0.0025}}$ | 0.83% | $0.0638_{\pm0.0024}$ | $\mathbf{0.0672_{\pm0.0017}}$ | 5.33% |

Table 9: Performance comparison on CSI800. The better results are indicated in boldface.

| Methods | CSI800 | | | | | |
|---|---|---|---|---|---|---|
|  | IC | | | RankIC | | |
|  | Original | Ours | *Improv.* | Original | Ours | *Improv.* |
| MLP | $0.0386_{\pm0.0023}$ | $\mathbf{0.0399_{\pm0.0006}}$ | 3.37% | $0.0450_{\pm0.0048}$ | $\mathbf{0.0467_{\pm0.0035}}$ | 3.78% |
| LSTM | $0.0377_{\pm0.0017}$ | $\mathbf{0.0412_{\pm0.0008}}$ | 9.28% | $\mathbf{0.0500_{\pm0.0030}}$ | $0.0494_{\pm0.0010}$ | -1.20% |
| GRU | $\mathbf{0.0380_{\pm0.0026}}$ | $0.0376_{\pm0.0010}$ | -1.05% | $0.0493_{\pm0.0030}$ | $\mathbf{0.0511_{\pm0.0011}}$ | 3.65% |
| SFM | $\mathbf{0.0385_{\pm0.0005}}$ | $0.0365_{\pm0.0015}$ | -5.19% | $0.0485_{\pm0.0011}$ | $\mathbf{0.0487_{\pm0.0022}}$ | 0.41% |
| GAT | $0.0379_{\pm0.0005}$ | $\mathbf{0.0397_{\pm0.0003}}$ | 4,75% | $0.0483_{\pm0.0009}$ | $0.0483_{\pm0.0009}$ | 0.00% |
| ALSTM | $0.0316_{\pm0.0031}$ | $\mathbf{0.0383_{\pm0.0013}}$ | 21.20% | $0.0418_{\pm0.0034}$ | $\mathbf{0.0492_{\pm0.0015}}$ | 17.70% |
| Transformer | $0.0423_{\pm0.0028}$ | $\mathbf{0.0426_{\pm0.0018}}$ | 0.71% | $\mathbf{0.0573_{\pm0.0016}}$ | $0.0556_{\pm0.0022}$ | -2.97% |

