# OpenReview forum: "Leveraging Diffusion Transformers for Stock Factor Augmentation in Financial Markets"
_ICLR.cc/2025/Conference — Submitted to ICLR 2025_

### Official Review · Reviewer_14Ku · 2024-10-17

**Soundness:** 3
**Presentation:** 3
**Contribution:** 3
**Rating:** 6
**Confidence:** 3

**Summary:**

This paper presents a diffusion transformer-based approach for performing data augmentation in financial markets, addressing the problem of low signal-to-noise ratio and data homogeneity, which can reduce model performance. The authors demonstrate that the proposed approach can improve the performance of various models, leading to improved returns.

**Strengths:**

- The paper is well-written, with appropriate motivation provided by the authors.

- Improvements in returns and Weighted-IC are observed by applying the proposed approach.

- Extensive evaluations are provided, demonstrating improvements across a wide range of architectures.

**Weaknesses:**

- Lack of comparison with other time-series augmentation approaches proposed in the literature. This is perhaps the most important concern. Even though, in my experience, performing meaningful augmentation on financial data is indeed very tricky, this should be demonstrated by evaluating a number of baseline approaches. Authors are encouraged to compare with recent related approaches, such as:

Kollovieh, Marcel, et al. "Predict, refine, synthesize: Self-guiding diffusion models for probabilistic time series forecasting." Advances in Neural Information Processing Systems 36 (2024).


Seyfi, Ali, Jean-Francois Rajotte, and Raymond Ng. "Generating multivariate time series with COmmon Source CoordInated GAN (COSCI-GAN)." Advances in neural information processing systems 35 (2022): 32777-32788.

Wiese, Magnus, et al. "Quant GANs: deep generation of financial time series." Quantitative Finance 20.9 (2020): 1419-1440.

Xia, Haochong, et al. "Market-GAN: Adding Control to Financial Market Data Generation with Semantic Context." Proceedings of the AAAI Conference on Artificial Intelligence. Vol. 38. No. 14. 2024.


- Traditional forecasting metrics (RMSE, MAPE, MAE) are not evaluated. Although measuring returns and demonstrating improvements are important, they do not provide the full picture. There is a complex interplay between the investment strategy and the actual forecasting capabilities of a model. The use of Weighted-IC does not fully address these concerns. Authors could include a table showing RMSE, MAPE, and MAE results alongside the existing metrics. This would provide a more comprehensive evaluation of the model's forecasting capabilities.

- The models are evaluated only on forecasting tasks. What about classification (e.g., direction prediction) and DRL approaches? Is the generated data expected to be useful in these cases as well? Authors could include experiments demonstrating the method's performance on classification/DRL tasks or approrately discuss in the limitations section whether they expect their approach to generalize to other types of financial prediction tasks and why/why not.

- Lack of qualitative results to demonstrate the ability to condition the generation process. It would be useful to provide some qualitative results to more intuitively understand the effect of the proposed approach. Authors could include a figure showing different timeseries to demonstrate the effects of the proposed approach.

**Questions:**

Please comment on the weaknesses noted.

---

> ### Author Response · Authors · 2024-11-21
> **Author Rebuttal**
>
> Dear Reviewer $\color{purple}{\text{14Ku}}$,
>
> Thanks for your time in reviewing our paper. Some of your comments really improve the quality of our paper. We believe the major concern might be due to the similarity and dissimilarity of **univariate and multivariate** time-series forecasting (generation).
>
> ---
>
> ### **Comment 1: Lack of comparison with other time-series augmentation approaches proposed in the literature.**
>
> Thanks for the constructive review. We've added Quant-GAN [1] , COSCI-GAN [2] and TimeVAE [3] as our baselines in Figure 8 in Line 514 ~ Line 518. Also for the rest of the method you suggested we also cite them as our important related work. The reason for the choice of the additional baselines are:
>
> (1) TSDiff [4] and Quant-GAN [1] are **univariate** time-series forecasting models and don't consider the correlations between multiple variables. Hence they are not suitable for **multivariate** generation tasks as ours. We choose to add one of them, Quant-GAN [1] , to be added to our baseline. We've tuned the parameter, however the result is even lower than the original probably due to the lack of correlation capturing.
>
> (2) COSCI-GAN [2] is a **multivariate** method. However, it doesn't generate corresponding label for the generated feature. Hence the result is not promising, either. Market-GAN [5] faces with the same problem, and needs more information such as i) long-term fundamentals, ii) mid-term market dynamics and iii) short-term history with volatility to run the task. As an alternate to [5], We add TimeVAE [3], a well-known **multivariate** method, as one of the baselines.
>
> Model | CSI300 | CSI800
> -------------------|------------------|------------------
> Original	| 0.2789±0.0376	| 0.0758±0.0307
> Quant-GAN	| 0.2595±0.0357	| 0.1186±0.0344
> COSCI-GAN	| 0.2640±0.0337	| 0.1208±0.0264
> Time-VAE	| 0.2875±0.0291	| 0.1155±0.249
> Ours	| **0.3127±0.0113**	| **0.1295±0.0292**
>
> From the baseline suggestions, we might guess that you're an expert in univariate time-series tasks. If you have some experiences or suggestions for applying these methods to multivariate methods, please tell us and we're always more than willing to make these methods better!
>
> ---
>
> ### **Comment 2: Additional Evaluation Metrics like MSE, RMSE and MAPE**
> We believe this suggestion is of great help to us.
>
> Since our label is the excess return ratio of a stock in 5 days, it is very likely that the label is close to zero. In this case, using MAPE (Mean Absolute Percentage Error) can be problematic:
> 1. Mathematical issue: MAPE is calculated as (|actual - predicted| / |actual|) * 100. When the actual value is very close to zero, this division can lead to extremely large or undefined percentage errors.
> 2. Interpretation problem: For values much smaller than 1, even small absolute errors can result in very large percentage errors, which can be misleading.
> 3. Asymmetry: MAPE treats positive and negative errors of the same magnitude differently when the actual values are small.
>
> Hence we didn't adopt MAPE as the metric. We've added a comparison about RMSE and MSE in Figure 6.
>
> Category | MSE(ori) | MSE(DDA) | RMSE(ori) | RMSE(DDA)
> -------------------|------------------|------------------|------------------|--------------
> ALL      | **0.0563** | 0.0586 | **0.2325** | 0.2389
> Top30   | 0.1117 | **0.1019** | 0.3226 | **0.3118**
> Top100 | 0.0799 | **0.0771** | 0.2763 | **0.2733**
> Tail100 | 0.0824| **0.0817** | 0.2788| **0.2787**
> Tail30 | 0.0995 | **0.0965** | 0.3046 | **0.3006**
>
> As can be seen from the table, the observation is consistent to IC and RankIC. Our model performs differently on different degrees of price fluctuation: it performs better on high-volatility stocks (top stocks and bottom stocks), however performs worse on low-volatility stocks (whose excess return is close to zero). We attribute the reason to the choice of target domain and source domain. The target domain consists of the largest 300 stocks in China A-share, these stocks are more established companies with stable earnings, hence tend to have lower volatility; the source domain consists of all of the stocks in China A-share,  which means the source domain have a higher volatility than the target domain. Hence distilling knowledge from source domain to target domain enhances the prediction ability of high-volatility stocks at the expense of the low-volatility stocks, and since our strategy is choosing top30 stocks, this property is promising and leads to higher profit. Figure 6 is around Line 438 and the discussion is around line 403.

---

> > ### Author Response · Authors · 2024-11-21
> > **Author Rebuttal (Continue)**
> >
> > ## **Comment 3: Will generated data be useful in classification and DRL task?**
> >
> > Thanks for the great question. We believe DiffsFormer is intended to address the **data scarcity** problem, and may be helpful to many problem or approaches may face with the sparse data problem, with modest changes. Hence we believe DiffsFormer could have a quite broad impact and applications.
> >
> > However, as for the two tasks you mentioned:
> >
> > (1) We believe direction prediction faces with the scarcity problem as we share the same input data type, hence Diffsformer may help with the direction prediction task from this perspective. However, since the label for direction prediction problem is binary, we may assume that it may have a higher demand for feature-label matching. Although DiffsFormer uses the original label for the generated feature which alleviates the problem and works for the regression task, we are not sure if it works for the classification task. We believe if a better strategy is developed to ensure the exact match of feature and label, our method would be able to help the classification problem.
> >
> > (2) We are not familiar with the Deep Reinforcement Learning approaches. We think this task requires expert knowledge in reward design, policy selection and rich experience in optimization to which we lack the capacity. Our strategy now is simple: a money-weighted position over the predicted top-30 stocks. We believe DRL in portfolio management may require the accurate prediction of the return of each stock where we can help, but currently it is not within our scope.
> >
> > We've discussed these two tasks in the limitation section in around Line 533. We would like to emphasize that our work offers valuable insights to the field, and our contribution is significant and substantial.
> >
> > ---
> >
> > ## **Comment 4: Lack of qualitative results to demonstrate the ability to condition the generation process.**
> > We fully agree that qualitative results are important to intuitively show the effect of the proposed approach. Due to page limit, we've arranged it to Figure 10 in Appendix E.1. We've thought about plotted different time-series to demonstrate the effects. However, our task is a cross-sectional generation task (multivariate time-series generation tasks with 158 dimensions) which is difficult to visualize ana analyze through different timeseries. Instead, we provide a qualitative result with t-SNE. Each sample in the plot is a projection of a 8*158 sequence. We can see from figure (a) that if we generate without editing, the distribution of the original and generated features deviate; however, from figure (c) if we edits the sample with conditioning and transfer learning methods, the augmented feature are generated by perturbing the original feature, hence two distributions are aligned.
> >
> > ---
> >
> > ### **Summary**
> > We are grateful for your positive evaluation of our motivation, presentation and effectiveness. We further provide the following responses:
> > - More baselines
> > - More test metrics
> > - Limitation discussion
> > - Qualitative result clarification
> > We hope our responses address your concerns. Overall, we believe our work makes good contributions, and we would appreciate your reconsideration. Thank you for your efforts again!
> >
> > ---
> >
> > [1] Wiese, Magnus, et al. "Quant GANs: deep generation of financial time series." Quantitative Finance 20.9 (2020): 1419-1440.
> >
> > [2] Seyfi, Ali, Jean-Francois Rajotte, and Raymond Ng. "Generating multivariate time series with COmmon Source CoordInated GAN (COSCI-GAN)." Advances in neural information processing systems 35 (2022): 32777-32788.
> >
> > [3] Desai et al. TimeVAE: A Variational Auto-Encoder for Multivariate Time Series Generation.
> >
> > [4] Kollovieh, Marcel, et al. "Predict, refine, synthesize: Self-guiding diffusion models for probabilistic time series forecasting." Advances in Neural Information Processing Systems 36 (2024).
> >
> > [5] Xia, Haochong, et al. "Market-GAN: Adding Control to Financial Market Data Generation with Semantic Context." Proceedings of the AAAI Conference on Artificial Intelligence. Vol. 38. No. 14. 2024.

---

> > > ### Comment · Reviewer_14Ku · 2024-11-23
> > >
> > > I would like to thank the authors for the detailed responses provides, which addressed to some degree some of the concerns I have raised. Therefore, I might raising my score accordingly.

---

> > > > ### Author Response · Authors · 2024-11-23
> > > >
> > > > Dear Reviewer $\color{purple}{\text{14Ku}}$,
> > > >
> > > > Thanks for raising your score! We're glad that we have addressed to some degree your concerns. Since there are still time till the rebuttal ends, if you're willing, we would be more than happy to resolve any remaining issues.
> > > >
> > > > We appreciate your professionalism, best regards,
> > > >
> > > > Submission 8851 authors

---

### Official Review · Reviewer_BbWb · 2024-11-06

**Soundness:** 3
**Presentation:** 1
**Contribution:** 2
**Rating:** 5
**Confidence:** 4

**Summary:**

The paper introduces a method, called DiffsFormer, using a diffusion model to generate stock data, addressing the overfitting issue in stock prediction tasks that arises from low signal-to-noise ratios and data homogeneity. Two techniques are used besides the standard diffusion model. The first one is to train the model on a large dataset, called source domain, with a large number of diffusion steps, then get augmented data points starting from a smaller dataset, called target domain, with a smaller number fiffusion steps. The second one is the predictor-free guidence technique to generate factors given the labels and other information, which is from (Ho & Salimans, 2022). Experments on CSI300 and CSI500 were conducted to show the advantage of the proposed method.

**Strengths:**

The concept of applying a diffusion model to generate stock data is interesting, especially regarding its potential to alleviate homogeneity within the original dataset. The experiments are well-designed, demonstrating the impact of data augmentation using the diffusion model on various prediction methods in financial metrics, and assessing the quality of the generated data through metrics measuring data fidelity and diversity. The results also show an improvement in return ratios with the diffusion model compared to previous data augmentation methods on stock data, displaying the potential for improving stock prediction methods using generative models.

**Weaknesses:**

1. The presentation of the paper is poor. It's hard to understand the methods and experiments. Some examples are listed below.

The figures are not referenced in order, i.e. some later figures are referenced first. Besides, some figures (Figs. 1 and 3) are not referenced at all, which is quite confusing.

The phrase “average number of stocks experiencing significant price drops” used to illustrate data homogeneity is confusing, and it is difficult to see how stocks within the same industry exhibit similar behavior based on the corresponding figure.

Which model was used to generate the results in Fig. 1? What does Fig. 1 exactly tell? Additionally, it shows data betwen Oct 2023 to Aug 2024, which does not correspond to the stated test sample period.

It's strange to claim in L237 "Since the target domain is a subset of the source domain... " considering that the relationship between the two domains hasn't been specified so far.

What is a data point like, a 158 dimensional factor vector for one day, or a sequence of factors for 8 days, say a 8*158 matrix?

In experiments, the source domains and target domains should be clearly staed in the main text instead of in Appendix because they are important information for understanding the results. How much more data was generated to augment the original dataset in experiments?

In L237, what does $\hat x_{T'}^{(t)}$ stand for? And in this line, why does only the last term in the x sequence have a hat symbol?

It's confusing to call the generation of a data point with the DM method as "editing" an existing data point. After reading the paper for several times I finally understood that this word is used in a metaphor. This word has caused much confusion to me.

Line 265: I didn't understand the following sentence for a long time: Since our labels are continuous rather than discrete, we refer to this mechanism as “predictor-free guidance.” After reading the reference (Ho, Salimans, 2022) I realized that this name follows the "classifier-free guidance" in that paper. But the present paper didn't make it clear.

What's CSIS in Table 4?

The metric values of Excess Return (ER) in Tables 1 and 2 are the same as the Return Ration (RR) in Tables 8 and 9. Please explain.

The authors are suggested to create a clear roadmap of the paper structure early on, ensure all figures are properly referenced and explained in order, and provide more explicit definitions of key concepts and variables used throughout.

2. The major technical contribution is training DM in the source domain with T diffusion steps, then get augmented data points in the target domain with T' diffusion steps, where T'<<T. I don't find experimental results directly supporting the advantage of this technique. It's suggested the authors show the results with different T'. This would demonstrate the impact and optimal setting of this key parameter. BTW, what values were used for T and T' for reporting the results in the paper?

3. The results are not as good as stated in abstract.

Tables 1&2 report weighted IC. It is true that weighted IC are more related to returns than IC and RankIC, but it is a more direct measurement of prediction accuracy since the prediction models are trained for predicting all stocks. Thus, it’s puzzling that many methods show lower IC after applying data augmentation with the diffusion model (Tables 8&9).

Many comparison results in Tables 1, 2, 8 and 9 may not be reliable, considering the large standard devision compared with the minor difference in average values. For instance, in table 2, on average Ours exceeds Original by 0.012, while the STDs of the two methods are 0.038 and 0.022. Though Ours improves Original by 11.96%, the difference may not be statistically significant. Many results like this are present in these tables. It's suggested to perform significance  test (e.g., t-tests or ANOVA) to report the comparison results.

The implementation of a “top30drop30” strategy for measuring excess returns raises concerns, as such strategies typically incur significant transaction costs in the A-share market. In our experience, such a strategy could never get positive returns on average. Please justify the choice of this strategy, provide results with transaction costs included, or implement and compare results using alternative strategies like the top-K approach.

**Questions:**

From Appendix E, it appears that IC and RankIC are actually worse for many prediction methods after applying the diffusion data augmentation. While it is true that accurately modeling tail stocks has little contribution to excess returns, it seems strange that so many methods exhibit lower IC and RankIC, given that the model is optimized for predicting all stocks. Please provide a more detailed analysis or explanation of why the proposed method improves excess returns while potentially decreasing overall prediction accuracy as measured by IC and RankIC.

How does T' influence the performance of the method?

Regarding the calculation of excess returns, were transaction costs included?

---

> ### Author Response · Authors · 2024-11-21
> **Author Rebuttal**
>
> Dear Reviewer $\color{green}{\text{BbWb}}$,
>
> We thank the reviewer for their great effort and the insightful comments! Please find our responses to each point below.
>
> ---
>
> ### **Comment 1: Typos and misunderstandings.**
>
> > Figure reference.
>
> Thanks for the suggestion. We overlook this issue: some figures are located in the Appendix but referenced in main text. We've changed it from Figure-reference to Section-reference in the revised version to make sure the figures are referenced in order. Also, we have referenced Figs 1. and 3 in Line 035 and Line 177 in the revised version to prevent from any confusion.
>
> > “average number of stocks experiencing significant price drops” is confusing.
>
> We agree with your comment that "average" is somehow confusing. We have modify the term "average number of stocks experiencing significant price drops" to "the total number of stocks in specific sector facing price drops" in around line 060.
>
> Figure 2(b) presents a cumulative representation of stock price movements, where the Y-axis denotes the number of stocks experiencing price declines in a given year, categorized by industry sector. The different colors in each bar represent various sectors, and the height of the color bar indicates the total number of stocks in specific sector facing price drops. The presence of substantial color blocks for specific sectors in certain years (e.g., larger blocks of blue, green and yellow in some years) suggests that when a sector is affected,
> it often impacts multiple stocks in that sector simultaneously. This sector-wise influence leads to similar behavior within the same industry and the decrease of effective number of samples.
>
> > Which model was used to generate the results in Fig. 1? What does Fig. 1 exactly tell? Additionally, it shows data betwen Oct 2023 to Aug 2024, which does not correspond to the stated test sample period.
>
> Fig. 1 is drawn by DiffsFormer (Diffusion-Augmented Transformer, the best model in our paper). The stock market is CSI300 Index. Actually Diffsformer has been deployed in April, 2023, for more than 1 years. Fig. 1 depicts the cumulative excess return and cumulative excess return with cost of our system. We can observe before 2023.4, the cumulative excess return is marginal, sometimes even negative for excess return with cost. After deploying our model, the cumulative excess return exhibits a progressive and substantial increase. We will release the code upon acceptance, and the reason why we report the back-test result during 2020~2022 is that we will release the diffusion model during this period rather than the latest one. We've revised the paper in around Line 036.
>
> > It's strange to claim in L237 "Since the target domain is a subset of the source domain... " considering that the relationship between the two domains hasn't been specified so far.
>
> We've missed a specific text discussion in the original version, with only Figure 4b clearly stating the relationship between source domain and target domain. To make it more clear, we'll explain the source domain and target domain. The CSI 300 comprise the largest 300 stocks traded on the Shanghai Stock Exchange and the Shenzhen Stock Exchange; CSI 800 is a larger dataset consisting of CSI 500 and CSI 300, aiming to add some stocks with smaller size. And CSIS means all stocks in the China A-share market. Hence CSIS = CSI800 + A-share stocks not contained in CSI800, CSI800 = CSI300 + CSI500. We further emphasize the relationship between target domain and source domain to avoid misunderstanding in blue script in around Line 245 in the revised version.
>
> > What is a data point like, a 158 dimensional factor vector for one day, or a sequence of factors for 8 days, say a 8*158 matrix?
>
> Thanks for the insightful question. A data point is a sequence of factors, a 8*158 matrix. We've clarified it in Line 185.
>
> > It's confusing to call the generation of a data point with the DM method as "editing" an existing data point. After reading the paper for several times I finally understood that this word is used in a metaphor. This word has caused much confusion to me.
>
> We're terribly sorry for the confusion made. Actually editing is not identical to generating. We don't sample from random noise, instead we perturb real samples and reverse to acquire new data points. As pointed out by Reviewer, this process is similar to SDEdit. The advantages of "edit" over "generating" are: (a) its capacity to distill new knowledge. By training diffusion model in source domain and "editing" in target domain, we aggregate both information from the target domain and source domain with weight controlled by the editing step T'. (b) Direct generating from noise can't decide the label of the generated feature, however editing from existing sample can somehow guarantee the generated feature and original sample share the same label, it is crucial for the training of downstream forecasting task. Hope that addresses your concern!

---

> > ### Author Response · Authors · 2024-11-21
> > **Author Rebuttal (Continue)**
> >
> > > Line 265: I didn't understand the following sentence for a long time: Since our labels are continuous rather than discrete, we refer to this mechanism as “predictor-free guidance.” After reading the reference (Ho, Salimans, 2022) I realized that this name follows the "classifier-free guidance" in that paper. But the present paper didn't make it clear.
> >
> > Sorry for not making the term clear. We've reformulated this sentence in line 296 as you suggested.
> >
> > > What's CSIS in Table 4?
> >
> > Thanks for the constructive question. CSIS are all of the stocks in China A-share market. We've added the concept in line 245 and line 477 in Table 4 Caption.
> >
> > > The metric values of Excess Return (ER) in Tables 1 and 2 are the same as the Return Ration (RR) in Tables 8 and 9. Please explain.
> >
> > Sorry for the confusions made. ER is the same as RR and it is a typo. As statistics in Tables are identical, we've deleted them in the Table 8 and 9 locating in Appendix to make it clear.
> >
> > > The authors are suggested to create a clear roadmap of the paper structure early on, ensure all figures are properly referenced and explained in order, and provide more explicit definitions of key concepts and variables used throughout.
> >
> > We've clearly added a roadmap in introduction, which help let readers know the contribution of the papers and where to find corresponding techniques near the end of the introduction in Line 087 ~ Line 104.
> >
> > ---
> >
> > ### **Comment 2: It's suggested the authors show the results with different T'. What values were used for T and T' for reporting the results in the paper?**
> >
> > We fully agree with you that T' is an important hyper-parameter in our work. Actually, we report the performance of different T' in Table 3, (a wrap table that may be overlooked, hence we've highlighted it with color). We choose T as 1000, following normal DM setting, and T' is chosen as 300 according to Table 3. BTW, a more detailed table is reported in Table 6 could help the reproduction of our work.
> >
> > Furthermore, besides T', we've added a summarization our contribution in Line 105:
> > - We reveal the importance of data augmentation in the context of stock forecasting and explore the use of diffusion stock transformer (DiffsFormer for short) to address data scarcity.
> >
> > - The framework integrates transfer learning to leverage knowledge from other markets, alleviating the difficulty of training DMs on sparse data. Additionally, the edit mechanism could obtain new features with original label with optimized efficiency, enabling training of the downstream forecasting task. For better alignment of the feature and the original label, we propose to employ excess return as the conditioning to enhance the relationship between them. Inspired by classfier-free guidance, a flexible predictor-free guidance approach is integrated as excess return is continuous rather than discrete.
> >
> > - We verify the effectiveness of DiffsFormer augmented training in CSI300 and CSI800 with nine commonly used machine learning models.
> >
> > ---
> >
> > ### **Comment 3: It’s puzzling that many methods show lower IC after applying data augmentation**
> >
> >  Category | MSE(ori) | MSE(DDA) | RMSE(ori) | RMSE(DDA)
> > -------------------|------------------|------------------|------------------|--------------
> > ALL      | **0.0563** | 0.0586 | **0.2325** | 0.2389
> > Top30   | 0.1117 | **0.1019** | 0.3226 | **0.3118**
> > Top100 | 0.0799 | **0.0771** | 0.2763 | **0.2733**
> > Tail100 | 0.0824| **0.0817** | 0.2788| **0.2787**
> > Tail30 | 0.0995 | **0.0965** | 0.3046 | **0.3006**
> >
> > As suggested by Reviewer $\color{purple}{\text{14Ku}}$, we added the experiment for MSE and RMSE in Figure 6 in Line 432 ~ Line 443. Upon review, we find MSE, RMSE have similar trends to IC and RankIC: our model performs differently **on different degrees of price fluctuation**: it performs better on high-volatility stocks (top stocks and bottom stocks), however performs worse on low-volatility stocks (whose excess return is close to zero). We attribute the reason to the choice of target domain and source domain. The target domain consists of the **largest** 300 stocks in China A-share, these stocks are more established companies with stable earnings, hence tend to have lower volatility; the source domain consists of all of the stocks in China A-share,  which means the source domain have a higher volatility than the target domain. Hence distilling knowledge from source domain to target domain enhances the prediction ability of high-volatility stocks at the expense of the low-volatility stocks, and since our strategy is choosing top30 stocks, this property is promising and leads to higher profit. This discussion has been added in Line 377 ~ Line 407.

---

> > > ### Author Response · Authors · 2024-11-21
> > > **Author Rebuttal (Continue)**
> > >
> > > ### **Comment 4: Significance Test**
> > > We appreciate your critical observation on the absence of significance testing in our initial analysis. We've applied T-test to assess the statistical significance of the differences observed between the performances in Table 1 and 2.
> > > p_value | CSI300 | CSI800
> > > --- | --- | ---
> > > MLP | 0.123 | 0.102
> > > LSTM | 0.868 | 0.758
> > > GRU | 0.157 | 0.0003
> > > SFM | 0.923 | 0.004
> > > GAT | 0.019 | 0.007
> > > ALSTM | 0.012 | 0.0005
> > > HIST | 0.249 | -
> > > Transformer | 0.016 | 0.280
> > >
> > > The result shows that your concern is right. We observe that most of the improvements are significant, while few of them are less significant or even not significant. In light of your feedback, we've incorporated these significance test outcomes into our manuscript.
> > >
> > > However, many prestigious works [1~3] have shown that low Signal-to-Noise Ratio may lead to high variance. Hence we believe it is unlikely to have a model who in the worst case is better than a worse model in the best case. We suppose a better model should: (1) have better average performance. (2) a lower standard variance. Our report in Table 1, 2 and Figure 8 has demonstrated this argument.
> > >
> > > As for real-world practical use, we can choose the model on a small validation dataset. We remain train dataset as the 2008.01\~2020.04, using 2020.04\~2020.12 to serve as validation dataset, and test on 2020.12\~2022.09.
> > > Performance | 300-ori | 300-aug | 800-ori | 800-aug
> > > --- | --- | --- | --- | ---
> > > MLP | 0.2278 | **0.2345** | **0.1292** | 0.1243
> > > LSTM | 0.2498 | **0.2587** | 0.1165 | **0.1408**
> > > GRU | **0.2167** | 0.2140 | 0.0828 | **0.1265**
> > > SFM | 0.2253 | **0.2289** | 0.0980 | **0.1415**
> > > GAT | 0.2333 | **0.3021** | 0.0849 | **0.0862**
> > > ALSTM | 0.2410 | **0.2757** | 0.0880 | **0.2257**
> > > HIST | **0.2420** | 0.2243 | - | -
> > > MTMD | 0.1408 | **0.1830** | - | -
> > > Transformer | 0.2688 | **0.3360** | 0.1583 | **0.2923**
> > >
> > > We've added significance test and best case comparision in Table 1 and Table 2 in the manuscript.
> > >
> > > ---
> > >
> > > ## **Comment 5: Trading Strategy**
> > >
> > > Thanks for the great question. Top30drop30 means that we keep the stocks with top30 predicted scores, and each stock will be droped if its score falls out of top30, regardless of its previous performance. Correct me if I'm wrong, this strategy is exactly the top-30 approach. The reason why we call it Top30drop30 is to compare it with Top30drop5 strategies where only the worst 5 stocks are sold and the best 5 are bought. As for the transaction costs, currently no transaction costs are included in all of the reported metrics. Figure 1 plots the transaction cost for our online model.
> > >
> > > ---
> > >
> > > ### **Summary**
> > > We are grateful for your positive evaluation of our motivation and effectiveness. We further provide the following responses:
> > > - Revise our manuscript to enhance reading experience.
> > > - In-depth analysis of falling IC and RankIC on low-volatility stocks.
> > > - Significance Test and Best case performance comparison
> > > We hope our responses address your concerns. Overall, we believe our work makes good contributions, and we would appreciate your reconsideration. Thank you for your efforts again!
> > >
> > > ---
> > >
> > > [1] Zhang et al. Understanding deep learning requires rethinking generalization. In ICLR'2017.
> > >
> > > [2] Michiaki Taniguchi and Volker Tresp. Averaging Regularized Estimators. In Neural Computation, 1997.
> > >
> > > [3] Zur et al. Noise injection for training artificial neural networks: A comparison with weight decay and early stopping. In Medical Physics, 2009.

---

> > > > ### Comment · Reviewer_BbWb · 2024-11-28
> > > > **Many problems remain**
> > > >
> > > > Thanks for the clarification. After the authors supplement and clarify many critical points such as CSIS data, I now understand the paper more. However, there are still many problems.
> > > > 1. Even after the authors' clarification about Fig 2b, I don't understand why counting the number of stocks in a specific sector (e.g., healthcare) facing price drops in a year indicates data homogeneity in that sector without knowing the total number of stocks in that sector. E.g., there are 100 stocks in the healthcare sector, and 20 of them facing price drops in a year, can we say the 100 stocks have much correlation in that year?
> > > > 2. In experiments, T was set to 1000. I don't find this information in the paper. Please note that this is important information because the authors claim that T' is much smaller than T, and the readers need to know what values of T' and T were used.
> > > > 3. This argument does not make sense: "distilling knowledge from source domain to target domain enhances the prediction ability of high-volatility stocks at the expense of the low-volatility stocks, and since our strategy is choosing top30 stocks, this property is promising and leads to higher profit." In fact, the models were tested on CSI 300, in which stocks have low-volatility. According to the author's argument above, the proposed method should have enhanced the prediction ability of models on all CSI stocks instead of only top 30 stocks.
> > > > 4. The significance test results in Tables 1&2  indicates that about half of the models obtained better results with the augmented data. It's good, but not good enough. The new results in the rebuttal make me confusing: it seems that the quality of results is highly dependent on the training/validation/testing set split. How can we trust the performance of the proposed method?
> > > > 5. Compared to Top30drop5 trading strategy, the top30-drop30 strategy will incur significant transaction cost, and in my experience, it is impossible to get positive reward in the A-share market. It's an unrealistic strategy.
> > > > 6. I think comparison using standard metrics such as IC, RankIC, RR makes more sense, instead of the weighted IC defined by the authors. According to IC and Rank IC values, the proposed method does not show clear improvement. Please note that according to the authors' argument about low volatility of CSI300 and CSI800, the proposed method should have obtained better results for all stocks in the two datasets instead of top stocks only. The weighted IC puts more emphasis on top stocks, which may not be much reasonable.
> > > > 7. The so-called editing is still unclear. Is the process in L243 from x_0^t to X_{T'}^t, or the process in L244 x_{T'}^t to \hat x_0^t called editing, or both? Please make it clear.
> > > > 8. L495-496: "When the source and target domains are identical, meaning no new information is introduced, DM still enhances performance." From Table 4, I don't see this conclusion. In fact, the improvements are marginal and I doubt if they can pass the significance test.
> > > > 9. On the two datasets CSI300 and CSI800 (Tables 1&2), why weren't the same set of models tested?
> > > > 10. The presentation still have some problems. E.g., several tables such as Tables 3&5  (and the corresponding main text) compare "Performance", but it's unclear which metric is used. BTW, in Table 3 W-Distance is used as a metric, but in the main text it is said that FID is in the table. Is FID identical to W-Distance? For another example, Figure 4b is referred before Figure 4a.

---

> > > > > ### Author Response · Authors · 2024-11-29
> > > > >
> > > > > Dear Reviewer $\color{green}{\text{Bbwb}}$,
> > > > >
> > > > > Thanks for your time effort and the detailed response which helps our work better. We'd like to address your concerns point by point:
> > > > >
> > > > > > Data Homogeneity
> > > > >
> > > > > Thanks for the great question. We agree that changing from **number** to **ratio** could make the argument clearer. As the pdf file can't be updated now, we will plot a new figure in the camera-ready version upon acceptance.
> > > > >
> > > > > Moreover, we argue that the current presentation provides **enough** information for data homogeneity. As discussed in Line 61: *"The presence of substantial color blocks for specific sectors in certain years (e.g., larger blocks of blue, green and yellow in some years) suggests that when a sector is affected, it often impacts multiple stocks in that sector simultaneously"*. **We are focusing on the sector-wide impact on the multiple (not necessarily all) stocks across year** which leads to the data homogeneity. Let's take healthcare you mentioned as an example: we observe substantial color blocks on healthcare in year 2011, 2018 and 2021. According to the official data (shenwan index), the healthcare sector fell by -30.89% in 2011; fell by -27.67% in 2018; fell by -5.73% in 2021; on the other hand, the color block is hard to see on healthcare in year 2009, 2010, where the healthcare sector increased by 102.79% and 29.69% respectively. We believe that it is a strong evidence of the existence of sector-wide impact.
> > > > >
> > > > > > the readers need to know what values of T' and T were used.
> > > > >
> > > > > Thanks for the advice. Diffusion step of 1000 is a typical choice for the diffusion model, and we didn't treat it as a hyper-parameter. We fully understand that readers from diverse backgrounds may benefit from additional context. As the pdf file can't be updated now, we will add a brief note in the camera-ready version upon acceptance.
> > > > >
> > > > > > About Low volatility and High volatility and Weighted-IC
> > > > >
> > > > > The reviewer correctly points out that CSI300 stocks have low-volatility, hence it is intuitive that CSIS (the whole A-share market) retains a higher volatility than CSI300. On top of that, "distilling from source domain (CSIS with higher volatility) to target domain (CSI300 with lower volatility)" would introduce more high-volatility information and knowledge to a low-volatility set, which enhances the prediction ability of (relative) high-volatility stocks within CSI300". However, the possible inherent gap between CSI300 and CSIS harms the (relative) low-volatility stocks within CSI300. That's why Diffsformer enhances the **relative** high-volatility stocks and harms the **relative** low-volatility stocks.
> > > > >
> > > > > As the CSI300 stocks tend to have low-volatility stocks, hence harming the **relative** low-volatility stocks may lead to a overall decreasing in IC and RankIC. However, enhancing the **relative** high-volatility stocks can improve the predicting ability on top stocks and bottom stocks, and bring about more excess returns since top stocks are more important during trading.
> > > > >
> > > > > > The new results in the rebuttal make me confusing: it seems that the quality of results is highly dependent on the training/validation/testing set split. How can we trust the performance of the proposed method?
> > > > >
> > > > > New experiment does **NOT** indicate that the quality of results is highly dependent on the training/validation/testing set split. Our new experiment is to prove that **our model in the best case is better than the baseline model in the best case**.
> > > > >
> > > > > Our previous experiment setting is to employ 2008.1 $\sim$ 2020.4 as the training set, and test on 2020.4 $\sim$ 2022.9, with no validation set involved due to data scarcity. To mitigate the randomness influence, we run experiments for 8 times and report the average and std of the result.
> > > > >
> > > > > Note that in noisy datasets, high variance is expected [1~3] and that perfect consistency across runs is unrealistic, hence we evaluate the trained models on a small validation set to choose the best model. Our aim is to compare between DiffsFormer and the baselines in best cases respectively. Since we don't have validation set before, we remain the previous training dataset, and split the original test dataset (2020.4 $\sim$ 2022.9) as validation dataset (2020.4 $\sim$ 2020.12) and new test dataset (2020.12 $\sim$ 2022.9), and choose the model which performs the best on validation dataset as the best model.
> > > > >
> > > > > ---
> > > > >
> > > > > [1] Zhang et al. Understanding deep learning requires rethinking generalization. In ICLR'2017.
> > > > >
> > > > > [2] Michiaki Taniguchi and Volker Tresp. Averaging Regularized Estimators. In Neural Computation, 1997.
> > > > >
> > > > > [3] Zur et al. Noise injection for training artificial neural networks: A comparison with weight decay and early stopping. In Medical Physics, 2009.

---

> > > > > > ### Author Response · Authors · 2024-11-29
> > > > > > **Official Comment by Authors (Part 2)**
> > > > > >
> > > > > > > Editing Process
> > > > > >
> > > > > > Our model takes $x_0^{(t)}$ (an existing sample in the target domain) as input, and outputs $\hat x_0^{(t)}$ (a new sample in the target domain that shows slight deviation from $x_0^{(t)}$). The whole generation process is called "editing".
> > > > > >
> > > > > > > On the two datasets CSI300 and CSI800 (Tables 1&2), why weren't the same set of models tested?
> > > > > >
> > > > > > The reason is that some models require additional information which we only have on CSI300. The explanation is located around Line 350, *"Note that HIST requires the concept of stocks to build the graph, therefore we don’t run it on CSI800 where the concepts are not available."* MTMD is a variation of HIST, therefore it requires this information, too.
> > > > > >
> > > > > > > Other typos and confusions.
> > > > > >
> > > > > > Thanks for the suggestions, as the pdf file can't be updated now, we will make corresponding changes in the camera-ready stage if accepted:
> > > > > > - We will change the "performance" to "excess return" in Table 3$\sim$5 in the revised version.
> > > > > > - Thanks for bringing "FID" and "W-Distance" to our attention. During rebuttal, we add some new content, and some content in the main text has been moved to appendix due to page limit. This is the cause for the confusion: In line 996, we explain the reason and state that "we directly calculate the Wasserstein Distance between generated and original data to serve as an alternate for FID". It is used to be in the Section 4.2 where we report Table 3. We'll move the explanation to the main text near Table 3 in the revised version.
> > > > > > - Thanks for pointing out the reference order about Figure 4, we will swaps the labels of the two figures: Make Figure 4a as Figure 4b and make Figure 4b as Figure 4a.
> > > > > >
> > > > > > ---
> > > > > >
> > > > > > Thanks for the detailed response again, we hope the discussion could address your concerns!
> > > > > >
> > > > > > Best regards,
> > > > > >
> > > > > > Submission 8851 Authors

---

> > > > > > > ### Author Response · Authors · 2024-12-03
> > > > > > > **Greatly appreciate your support!**
> > > > > > >
> > > > > > > Dear Reviewer $\color{green}\text{Bbwb}$,
> > > > > > >
> > > > > > > We are extremely grateful for the insightful suggestions to make our work better!
> > > > > > >
> > > > > > > We've tried our best to address your concerns. As the discussion phase comes to a close, if our response has solved most of your concerns, we humbly ask if you might consider raising your score.
> > > > > > >
> > > > > > > If you believe there are still areas in our paper that could be further improved, we would be more than happy to engage in any additional discussion to address any remaining concerns, particularly as the rebuttal period is about to conclude in just a few hours.
> > > > > > >
> > > > > > >
> > > > > > > Thank you once again for your thoughtful engagement! Your support at this stage is immensely important to us!
> > > > > > >
> > > > > > > Best regards,
> > > > > > >
> > > > > > > Submission 8851 Authors

---

### Official Review · Reviewer_vZrE · 2024-11-10

**Soundness:** 2
**Presentation:** 1
**Contribution:** 1
**Rating:** 3
**Confidence:** 3

**Summary:**

The paper proposes a framework for increasing data by augmentation due to shortage of data in stock market prediction task. The authors propose to incorporate some target domain information while increase variation in augmented source domain data. Authors did so by, what they call, ‘editing’, which is essentially using diffusion models’ inherent properties to start from noisy version of target domain data. Authors also incorporate conditional information to simulate data by industry/sector.

**Strengths:**

The problem tackled in the paper is quite attractive indeed. Stock market prediction is one of the most financially lucrative applications of ML.

**Weaknesses:**

Note #1: I was assigned as an emergency reviewer and hence had limited time reading the paper. My assessment is based on evaluating the overall idea.

Note #2: I have little expertise on the specific application domain (stock market prediction) but have exposure to general generative modelling.

The paper, although tackles a lucrative problem, is in no way, novel. The paper mostly uses existing and well-known ideas and applies to a specific problem domain. Also, the writing quality is quite below the bar.

1. The (lack of) novelty is a big factor in my assessment of the paper. The paper uses Diffusion Models, which are not particularly known to be good for sequential data. There is not much discussion about why diffusion model or transformer architecture was chosen. The specific concept of ‘editing’ that was presented in the paper is quite a well-known idea called [SDEdit](https://arxiv.org/abs/2108.01073) which exists for quite a while. The authors did not cite or discuss it.
2. The time efficiency improvement proposed in section 3.2 is rather unnecessary and can be incorrect. The training complexity does not depend on T enhance it makes no difference to reduce the range of time during training. It can in fact lead to a poor model if one does not train it along the entire time horizon.
3. The idea of guidance in diffusion is also well known and the authors only seem to have used it as a black-box.

Moreover, the writing quality of the paper is below the bar. Technical concepts of diffusion models are not very well written or in some cases misleading or incorrect. The specifics of the application that is the variables and other quantities related to stock market prediction isn't very well explained.  The terms like stock factors in return ratio are used throughout the introduction section even though they are defined in section 2. This hinders the reading experience. The terms like SNR and data homogeneity are vaguely defined only in text. They should have been properly defined in mathematical terms.

Lastly, I feel like this paper being very specific in its application should be submitted in a domain specific conference and perhaps not a very good fit for ICLR.

**Questions:**

No further questions please see the witness section.

---

> ### Author Response · Authors · 2024-11-21
> **Author Rebuttal**
>
> Dear Reviewer $\color{orange}{\text{vZrE}}$,
>
> Thanks for the comments. To address your concerns, we provide detailed responses below.
>
> ---
>
> ### **Comment 1: The paper uses Diffusion Models, which are not particularly known to be good for sequential data. There is not much discussion about why diffusion model or transformer architecture was chosen.**
>
> Thanks for the question. Actually, besides computer vision, diffusion models has been applied to sequential data widely, including but not restricted to CSDI [1], TimeGrad [2],  D^3VAE [3],  DSPD [4], TSGM [5], WaveGrad [6]. Most of these prominent papers published at top conferences and journals as ICML, ICLR, NeurIPS, TMLR, and has received great citations demonstrating their attraction to the wide community. Most of the papers have replaced UNET within the diffusion model with Transformer, as Transformers have strong abilities in handling sequential data. We mistakenly suppose it is intuitive and fails to discuss in our previous version. **We've stated it in the revised version in around Line 70**. Furthermore, none of the previous model are applied to financial domain, and they primarily focus on sequence imputation and sequence forecasting, rather than sequence generation.
>
> ---
>
> ### **Comment 2: the well-known idea called SDEdit which exists for quite a while.**
>
> Thanks for bringing this to our attention. We regret overlooking this important work in our initial submission. **We've added a discussion part in Line 256 ~ Line 262.**
>
> Upon review, we note several similarities:
>
> - SDE indeed serves as the theoretical support for both of the problems
> - the perturbing and reverse process looks alike.
>
>
> However our approach differs in the goal:
>
> - In our opinion, the goal of SDEdit is to generate both faithful and realistic image given input guidance image; As for diffsformer, it transfers the knowledge from a larger source domain which helps generate sequences in the test domain: By training diffusion model in source domain during training stage and starting from real data in target domain during inference, we aggregate both information from the target domain and source domain, and the aggregation contribution could be controlled by the editing step T'.
> - Downstream forecasting task with transformer is a supervised-learning task, hence we need to provide the label for the generated data. "Edit" could support that the generated data and the original label matches.
>
> It is important to note that, the core novelty is the **knowledge transfer to alleviate the low signal-noise-ratio problem with the label unchanged**. We notice that training on a source domain and testing on the test domain leads to performance drop, and diffusion models capturing the data distribution could act well as a data augmentation tool. We believe our contribution is significant, we've added analysis in introduction around  Line 087 ~ Line 095; the empirical support is in Table 4, around Line 481.
>
> ---
>
> ### **Comment 3: The idea of guidance in diffusion is well known**
> Thanks for the point. Stock forecasting task with transformer is a supervised-learning task. However, traditional DM doesn't provide label for the generated feature. Concatenate a label dimension at the end of the feature won't work (Figure 10 and Table 7), **that's the true reason we propose to employ ground truth (e.g., excess return) as the conditioning**. We incorporate conditioned DM and treat the generated data and the original label as the parallel data to train the downstream model and get remarkable improvement. We believe discovering and formulating this problem and the attempt to solve the problem with well-established methods is valuable, even though the classifier-free guidance is already a mature method with solid theoretical foundation.
>
> ---
>
> [1] Tashiro et al. CSDI: Conditional Score-based Diffusion Models for Probabilistic Time Series Imputation. In NeurIPS'21.
>
> [2] Rasul et al. Autoregressive denoising diffusion models for multivariate probabilistic time series forecasting. In ICML'21.
>
> [3] Li et al. Generative time series forecasting with diffusion, denoise, and disentanglement. In NeurIPS'22.
>
> [4] Bilos et al. Modeling Temporal Data as Continuous Functions with Stochastic Process Diffusion. In ICML'23.
>
> [5] Nikitin et al. TSGM: A Flexible Framework for Generative Modeling of Synthetic Time Series. In NeurIPS'24.
>
> [6] WaveGrad: Estimating Gradients for Waveform Generation. In ICLR'21.

---

> > ### Author Response · Authors · 2024-11-21
> > **Author Rebuttal (Continue)**
> >
> > ### **Comment 4: The time efficiency improvement is rather unnecessary and can be incorrect.**
> > Thanks for the question, and we believe there is a misunderstanding.
> >
> > If we define T as the total diffusion step during forward process, the training Loop for diffusion models could be like:
> > - Sample real data $x_{0}$ from the training dataset
> > - Sample a random timestep t from {0, 1, ..., T-1} or {1, ..., T}
> > - Add noise to $x_0$ to create $x_t$
> > - Pass $x_t$ and t through the neural network
> > - Compute loss between predicted and actual noise
> >
> > As you mentioned, the length of the entire time horizion matters, as it affects the value of $\alpha_t$ and $\beta_t$. Hence in our method, we remain T=1000 and initialize the $\alpha_t$ and $\beta_t$.
> > Since we only perturb seed point for T' << T steps then reverse, we believe $p_{\theta}(x_{t−1}|x_{t})$ will not be used for t larger than T'. In our paper, we set T'= 300.
> > From the training loop, we could observe the probability of choosing specific t < T is 1/999, and choosing specific t < T' is 1/299. In other words, for the same training epochs, the probability of selecting useful timestep increased to **more than three times** the original after decreasing the candidate set size from T to T'. Although theoretical time complexity doesn't change, the convergence speed becomes faster and real training time decreases. Figure 9 reflects the phenomenon. We've added more clarification in Section 3.2 in Line 284. Hope that the misunderstanding is mitigated!
> >
> > ---------
> >
> > ### **Comments 5:**
> > ### **- Technical concepts of DM are not very well written or in some cases misleading or incorrect;**
> >
> > ### **- the variables and other quantities related to stock market prediction isn't very well explained;**
> >
> > ### **- stock factors in return ratio are used throughout the introduction section even though they are defined in section 2;**
> >
> > ### **- The terms like SNR and data homogeneity should have been properly defined in mathematical terms.**
> >
> > We greatly appreciate your observations and would be most grateful if you could kindly provide some specific examples to help us address your concerns more effectively:
> > - Regarding the technical concepts of diffusion models, most of the papers follow the same formulation. There might be some typos in our equations that makes you feel wrong. Could you please point out which particular sections or statements you found potentially inaccurate? This would be immensely helpful in our revision process.
> > - In around Line 182 Diffusion Process, we've introduced the variables and other quantities about input data. Besides, we've used almost one page to introduce the background of the task. If there are more specific variables you feel require more clarification, please indicate and we're more than happy to explain them in the manuscript.
> > - We apologize for any confusion caused by the use of terms like "stock factors" and "return ratio" in the introduction. We have given text explanations and refer readers who are unfamiliar to the concepts to Section 2 in Line 36 and Line 47.
> > - Your point about SNR is well-taken. We've defined SNR in mathematical terms in Section 2 in Line 125 ~ Line 128. As for data homogeneity, it is a qualitative concept rather than a quantitative one. Many metrics such as variance, entropy, cosine similarity could serve as an indicator of the homogeneity property.
> >
> > ---------
> >
> > ### **Comments 6: Lastly, I feel like this paper being very specific in its application should be submitted in a domain specific conference and perhaps not a very good fit for ICLR.**
> > We acknowledge that our work is a economic-specific application paper. And we believe that our paper is a good fit for domain specific conference. However, we would like to kindly remind that:
> >
> > In call for paper page (https://iclr.cc/Conferences/2025/CallForPapers), the subject area for ICLR 2025 is: *"We consider a broad range of subject areas including **feature learning**, metric learning, compositional modeling, structured prediction, reinforcement learning, uncertainty quantification and issues regarding large-scale learning and non-convex optimization, as well as **applications** in vision, audio, speech, language, music, robotics, games, healthcare, biology, sustainability, **economics**, ethical considerations in ML, and others."*
> >
> > As our paper aims to do feature learning (data augmentation) from a representation learning perspective, we believe our paper is a good fit for ICLR.
> >
> > ---
> >
> > ### **Summary**
> > We are grateful for your positive evaluation of our motivation. We further provide the following responses:
> > - An in-depth discussion about novelty.
> > - Revise our manuscript to enhance reading experience.
> >
> > We hope our responses address your concerns. Overall, we believe our work makes good contributions, and we would appreciate your reconsideration. Thank you for your efforts again!

---

> > > ### Author Response · Authors · 2024-12-03
> > > **Greatly appreciate your support!**
> > >
> > > Dear Reviewer $\color{orange}\text{vZrE}$,
> > >
> > > We are extremely grateful for the insightful suggestions to make our work better!
> > >
> > > We've tried our best to address your concerns. As the discussion phase comes to a close, if our response has solved most of your concerns, we humbly ask if you might consider raising your score.
> > >
> > > If you believe there are still areas in our paper that could be further improved, we would be more than happy to engage in any additional discussion to address any remaining concerns, particularly as the rebuttal period is about to conclude in just a few hours.
> > >
> > > Thank you once again for your thoughtful engagement! Your support at this stage is immensely important to us!
> > >
> > > Best regards,
> > >
> > > Submission 8851 Authors

---

### Official Review · Reviewer_ZDKm · 2024-11-10

**Soundness:** 2
**Presentation:** 3
**Contribution:** 3
**Rating:** 5
**Confidence:** 4

**Summary:**

The paper presents a novel approach, DiffsFormer, to address data scarcity in stock forecasting by leveraging diffusion-based data augmentation combined with transformer models. This method aims to mitigate the common challenges of low signal-to-noise ratio (SNR) and data homogeneity in stock data, thus improving predictive model performance.

**Strengths:**

+ The novel use of diffusion models for augmenting stock forecasting data is somehow technically sound.
+ Clear and robust experimental validation on real-world financial datasets.
+ Good writing and organization.

**Weaknesses:**

+ The approach uses transfer learning to train the diffusion model on a large source domain (i.e., broader stock market data) and applies it to a smaller target domain (e.g., CSI300 or CSI800). However, this raises the question of how domain differences (e.g., market structure, regulations, or trading behavior) may affect the generalizability of the learned knowledge. How does the model handle potential domain shift between source and target domains, particularly when the characteristics of the two datasets are fundamentally different (e.g., emerging vs. developed markets)?
+  The paper discusses the trade-off between data fidelity (keeping the data close to the original domain) and diversity (introducing variability in the data). Is there any risk of generating overfitted data that aligns too closely with the source domain but does not generalize well to the target domain? Could the approach lead to a loss of diversity in the generated data?
+ While excess return is a critical performance metric in financial markets, it may not fully capture the model’s stability or generalization ability, which are also important for real-world applications.
+ Diffusion models, especially when combined with complex transformers, are inherently difficult to interpret.  Could the authors explore feature importance or other interpretability techniques (e.g., SHAP, LIME) to provide insights into which factors most influence the model's decisions?
+ The paper compares DiffsFormer against multiple baseline models (e.g., LSTM, GRU, Transformer) and shows significant improvements. However, it is unclear if the comparison includes the latest state-of-the-art models in the domain of stock forecasting?
+  Are there alternative augmentation strategies, like generative adversarial networks (GANs), that could also serve as a competitive baseline?

**Questions:**

Check Weaknesses.

---

> ### Author Response · Authors · 2024-11-21
> **Author Rebuttal**
>
> Dear Reviewer $\color{blue}{\text{ZDKm}}$,
>
> We appreciate your comments! We believe the major concern might be due to a bit of misunderstanding on the relationship between source domain and target domain. We have dedicated to revising the manuscript to address your concerns.
>
> ---
>
> ### **Comments 1, 2 and 3: Model's stability and Generalization ability**
> (1) **Generalization of the knowledge**. We fully understand your concern and indeed care about the generalizability of the learned knowledge. To tackle the problem, we make our best effort to make the source domain and the target domain have the similar distribution. Our choice of source domain is a **superset** of the target domain, i.e. every sample in the target domain is seen in the target domain, which helps alleviate the out-of-distribution problem since the characteristics of the two datasets will NOT be fundamentally different. To make it more clear, we'll explain the source domain and target domain. The CSI300 comprise 300 stocks in the China A-share market; CSI800 is a larger dataset consisting of CSI500 and CSI300, aiming to add some stocks to CSI300. And CSIS means all stocks in the China A-share market. Hence **CSIS = CSI800 + A-share stocks not contained in CSI800, CSI800 = CSI300 + CSI500**. We're not transferring knowledge from a developed market such as NASDAQ to emerging market like China A-share, which we also think takes the risk of out-of-distribution. We further emphasize the relationship between target domain and source domain to avoid misunderstanding in the revised version (Line 244 ~ Line 249). Since the two domains don't deviate too much, it is not necessary to worry much about the generalization ability.
>
> (2) **Overfitting**. We fully agree that sparse data may lead to overfitting issues in DM training. Accordingly, in Section 3.4 we propose a novel loss-guided diffusion mechanism which introduces stronger noise to easily fitted data points, results from Figure 7 shows the effectiveness of this mechanism in increasing information ratio of the model, which is an indicator of generalization ability and stability.
>
> (3) **Diversity**. We believe data diversity is a quite important metric controlled by the guidance strength of the DM. As you mentioned, we find the diversity of the data decreases when the guidance strength is high (i.e. 4 or 5), the model performance drops due to the decrease of diversity. On top of that, we recommend a moderate guidance strength where the diversity of the generated data could be ensured.
>
> ---
>
> ### **Comment 3: excess return may not fully capture the model’s stability or generalization ability**
> We fully agree with this comment that excess return ratio is only used to evaluate the profitability of a model. In finance, the stability and generalization ability of the model is often reflected by another metric, information ratio. We've reported it in Figure 7 in around Line 428. Information ratio measures the excess return of an investment relative to its benchmark, adjusted for the risk taken to achieve that excess return.
>
> $IR = (R_p - R_b) / TE$
>
> Where $R_p$= Return of the portfolio or strategy;
> $R_b$= Return of the benchmark;
> TE = Tracking Error (standard deviation of the difference between portfolio and benchmark returns)
>
> **Here is some explanation of the metric:**
> 1. The numerator $R_p-R_b$ is the excess return, also known as alpha. It shows how much the portfolio or strategy outperformed (or underperformed) the benchmark.
> 2. The denominator (TE) is the tracking error, which measures the consistency of the excess returns. It's calculated as the standard deviation of the excess returns over time.
> 3. The information ratio (IR) measures portfolio returns above the returns of a benchmark, against the **volatility of those returns**.
>
> ---
>
> ### **Comments 4: Feature Importance**
>
> Thanks for your suggestion. We fully agree that feature importance is of great helpful to model interpretability. In Figure 1(a), we plot the Pearson Coefficient between the label and each feature dimension, and find most of the absolute values are below 0.03, which indicates that **feature and label has very weak correlations and isolation feature may not reflect its real role in the multi-factor interplay**. Following your suggestions, **we implement LIME to visualize the feature importance with the last time step in the sequence**, and find most of the values are equal to zero. It indicates that some intuitive methods could not visualize the feature importance, and we're always willing to explore more complex method to capture non-linear and context-dependent nature, as well as the temporal dependencies in DiffsFormer.

---

> > ### Author Response · Authors · 2024-11-21
> > **Author Rebuttal (Continue)**
> >
> > ### **Comments 5: latest state-of-the-art models in the domain of stock forecasting**
> > Thanks for the suggestion. HIST [1] is one of the SOTA model that submitted their codes to https://github.com/microsoft/qlib. Moreover, we have added a comparison with the MTMD [2] model. Our current comparison was carefully conducted within the same time frame in the paper. The new results has been added to Table 1, 2, and 8.
> >
> > Model | ER | WeightedIC
> > -------------------|------------------|------------------
> > HIST | 0.2272±0.0352 | 0.0249±0.0066
> > HIST(Ours) | **0.2410±0.0207**  | **0.0317±0.0026**
> > MTMD | 0.2129±0.0355  | 0.0316±0.0027
> > MTMD(Ours) | **0.2547±0.0207** | **0.0347±0.0021**
> >
> > -----
> >
> > ### **Comments 6: Alternative augmentation strategies serve as a competitive baseline**
> > Thanks for the advice. As suggested by Reviewer $\color{purple}{\text{14Ku}}$, we have added [3~5] as our baselines to the paper, please refer to Figure 8 around Line 490.
> >
> > Model | CSI300 | CSI800
> > -------------------|------------------|------------------
> > Original	| 0.2789±0.0376	| 0.0758±0.0307
> > Quant-GAN	| 0.2595±0.0357	| 0.1186±0.0344
> > COSCI-GAN	| 0.2640±0.0337	| 0.1208±0.0264
> > Time-VAE	| 0.2875±0.0291	| 0.1155±0.249
> > Ours	| **0.3127±0.0113**	| **0.1295±0.0292**
> >
> > However, we find these baselines:
> >
> > (1) cannot generate (feature, label) pairs. If original labels serve as the supervison signal for the generated feature, the performance drops. (2) cannot generate multivariate time-series data. Our cross-sectional data contains 158 factors, if we treat this multivariate sequence as 158 univariate sequences and generate them separately, the performance will face a significant drop since we overlook the correlation between features.
> >
> > ---
> >
> > ### **Summary**
> > We are grateful for your positive evaluation of our presentation, novelty, and effectiveness. We further provide the following responses:
> > - An in-depth discussion about generalization and Overfitting.
> > - Additional experiments in stock forecasting and data augmentation.
> > - Feature Importance.
> > We hope our responses address your concerns. Overall, we believe our work makes good contributions, and we would appreciate your reconsideration. Thank you for your efforts again!
> >
> > ---
> >
> > [1] Xu et al. HIST: A Graph-based Framework for Stock Trend Forecasting via Mining Concept-Oriented Shared Information.
> >
> > [2] Wang et al. MTMD: Multi-Scale Temporal Memory Learning and Efficient Debiasing Framework for Stock Trend Forecasting.
> >
> > [3] Seyfi, Ali, Jean-Francois Rajotte, and Raymond Ng. "Generating multivariate time series with COmmon Source CoordInated GAN (COSCI-GAN)." Advances in neural information processing systems 35 (2022): 32777-32788.
> >
> > [4] Wiese, Magnus, et al. "Quant GANs: deep generation of financial time series." Quantitative Finance 20.9 (2020): 1419-1440.
> >
> > [5] Desai et al. TimeVAE: A Variational Auto-Encoder for Multivariate Time Series Generation.

---

> > > ### Comment · Reviewer_ZDKm · 2024-11-26
> > >
> > > Thanks for your responses. I keep my score as 5.

---

> ### Author Response · Authors · 2024-11-27
>
> Dear Reviewer $\color{blue}{\text{ZDKm}}$,
>
> Thanks for the response. As the discussion period has been extended, we believe that we still have time to address your concerns. Do you still have any concerns? Please feel free to tell us and we are always more than happy to address them.
>
> Best regards,
>
> Submission 8851 Authors

---

> > ### Author Response · Authors · 2024-11-27
> > **Clarification for LIME on feature importance**
> >
> > Dear Reviewer $\color{blue}{\text{ZDKm}}$,
> >
> > We have carefully checked our rebuttal. We guess you might be concerned about our answers about **feature importance discussion about LIME**. We hereby provide with the full LIME result and the **explanation for the top-5 feature** we got.
> >
> > 101_t-7 <= -0.34: 0.030
> >
> > 156_t-7 <= -0.70: 0.021
> >
> > 146_t-7 <= -0.70: -0.020
> >
> > 137_t-7 > 0.90: 0.019
> >
> > 11_t-6 <= -0.94: -0.018
> >
> > 0 < 11_t-7 <= 0.56: 0.016
> >
> > 67_t-7 <= -0.66: -0.015
> >
> > 43_t-7 <= -0.57: 0.015
> >
> > 101_t-6 <= -0.34: 0.014
> >
> > 132_t-7 > 0.79: -0.013
> >
> > 29_t-7 <= -0.67: 0.013
> >
> > 100_t-6 <= -0.67: 0.013
> >
> > 14_t-7 > 0.66: -0.013
> >
> > 151_t-7 > 0.70: -0.012
> >
> > 152_t-7 > 0.72: 0.012
> >
> > 9_t-5 <= -0.68: 0.012
> >
> > 52_t-7 > 0.6: 0.012
> >
> > 100_t-7 <= -0.67: 0.012
> >
> > 26_t-7 <= -0.58: 0.012
> >
> > 140_t-7 > 0.83: 0.012
> >
> > 110_t-6 <= -0.67: 0.012
> >
> > 10_t-1 <= -0.57: -0.011
> >
> > 52_t-6 > 0.6: 0.011
> >
> > 110_t-7 <= -0.67: 0.011
> >
> > 5_t-6 <= -0.57: -0.011
> >
> > 59_t-5 <= -0.92: -0.011
> >
> > 156_t-6 > 0.64: -0.011
> >
> > -0.57 < 5_t-7 <= 0: -0.011
> >
> > 51_t-7 > 0.58: 0.011
> >
> > 8_t-7 > 0.64: -0.010
> >
> > 0 < 98_t-7 <= 0.67: 0.010
> >
> > -0.54 < 112_t-0 <= 0: 0.010
> >
> > 26_t-6 <= -0.58: 0.010
> >
> > 81_t-7 > 0.81: -0.010
> >
> > 101_t-5 <= -0.34: 0.010
> >
> > 146_t-5 > 0.63: 0.010
> >
> > 0 < 78_t-7 <= 0.67: 0.010
> >
> > 49_t-7 > 0.58: 0.009
> >
> > 0.01 < 49_t-4 <= 0.59: 0.009
> >
> > 42_t-5 > 0.65: -0.009
> >
> > 26_t-5 <= -0.58: 0.009
> >
> > 97_t-6 > 0.62: 0.009
> >
> > 141_t-6 > 0.85: -0.009
> >
> > 9_t-6 <= -0.68: 0.009
> >
> > 70_t-7 <= -0.66: -0.009
> >
> > 119_t-7 > 0.69: 0.009
> >
> > 61_t-7 > 0.54: 0.009
> >
> > 96_t-7 > 0.61: 0.009
> >
> > 58_t-5 <= -1.04: 0.009
> >
> > 134_t-6 <= -0.57: -0.009
> >
> > 95_t-6 > 0.6: -0.008
> >
> > 111_t-7 <= -0.67: 0.008
> >
> > 138_t-6 > 0.74: 0.008
> >
> > -0.71 < 156_t-3 <= -0.01: 0.008
> >
> > 84_t-7 <= -0.67: 0.008
> >
> > -0.67 < 110_t-0 <= 0: 0.008
> >
> > 124_t-7 <= -0.69: -0.008
> >
> > 153_t-0 <= -0.73: -0.008
> >
> > 3_t-7 <= -0.57: 0.008
> >
> > 134_t-5 <= -0.57: -0.008
> >
> > 26_t-2 <= -0.58: 0.008
> >
> > 0 < 110_t-2 <= 1.01: -0.008
> >
> > 139_t-7 > 0.79: 0.008
> >
> > 102_t-7 <= -0.67: -0.008
> >
> > 156_t-5 > 0.63: -0.007
> >
> > 0_t-6 > 0.69: 0.007
> >
> > 0 < 57_t-0 <= 0.91: -0.007
> >
> > 108_t-2 > 0.67: 0.007
> >
> > -0.01 < 126_t-2 <= 0.64: -0.007
> >
> > 39_t-7 > 0.65: 0.007
> >
> > 10_t-5 <= -0.57: -0.007
> >
> > -0.01 < 37_t-2 <= 0.61: -0.007
> >
> > 143_t-6 > 0.63: -0.007
> >
> > -0.61 < 4_t-5 <= 0: 0.007
> >
> > 9_t-4 > 0.63: -0.007
> >
> > -0.61 < 15_t-1 <= 0: 0.007
> >
> > 12_t-5 <= -0.64: -0.007
> >
> > 141_t-7 > 0.85: -0.007
> >
> > 100_t-4 <= -0.67: 0.007
> >
> > 10_t-7 <= -0.57: -0.007
> >
> > -0.69 < 116_t-7 <= -0.01: 0.007
> >
> > 8_t-0 <= -0.69: 0.007
> >
> > -0.74 < 36_t-3 <= -0.01: 0.007
> >
> > 6_t-7 > 0.80: -0.007
> >
> > 146_t-6 > 0.64: 0.007
> >
> > 123_t-5 > 0.64: -0.007
> >
> > 9_t-3 <= -0.68: 0.007
> >
> > 86_t-7 <= -0.6: -0.007
> >
> > 11_t-5 <= -0.94: -0.007
> >
> > 0 < 140_t-2 <= 0.82: 0.007
> >
> > -0.69 < 123_t-7 <= 0: 0.007
> >
> > -0.67 < 106_t-5 <= 0: -0.007
> >
> > 74_t-7 <= -0.67: -0.007
> >
> > -0.6 < 4_t-2 <= 0: 0.007
> >
> > 0 < 131_t-7 <= 0.76: 0.006
> >
> > 0 < 112_t-3 <= 0.67: -0.006
> >
> > 0 < 6_t-5 <= 0.8: 0.006
> >
> > 0 < 64_t-4 <= 0.9: -0.006
> >
> > 81_t-4 > 0.81: -0.006
> >
> > 25_t-0 <= -0.59: -0.006
> >
> > 43_t-6 <= -0.57: 0.006
> >
> > 109_t-3 > 0.67: 0.006
> >
> > 129_t-0 > 0.74: -0.006
> >
> > 155_t-4 > 0.63: 0.006
> >
> > -0.01 < 37_t-7 <= 0.61: 0.006
> >
> > 27_t-4 <= -0.58: 0.006
> >
> > 0 < 136_t-0 <= 0.91: -0.006
> >
> > 0 < 99_t-4 <= 0.67: 0.006
> >
> > -0.86 < 60_t-0 <= 0.01: 0.006
> >
> > 99_t-2 > 0.67: 0.006
> >
> > 83_t-7 <= -0.67: -0.006
> >
> > 156_t-4 > 0.63: -0.006
> >
> > -0.56 < 47_t-6 <= 0: -0.006
> >
> > -0.63 < 148_t-7 <= 0: 0.006
> >
> > 85_t-1 <= -0.67: -0.005
> >
> > 1_t-0 <= -0.59: -0.005
> >
> > 130_t-4 <= -0.6: 0.005
> >
> > -0.67 < 78_t-5 <= 0: -0.005
> >
> > -0.54 < 112_t-6 <= 0: 0.005
> >
> > 1_t-6 > 0.86: 0.005
> >
> > 96_t-6 > 0.61: 0.005
> >
> > -0.69 < 113_t-7 <= 0: -0.005
> >
> > 131_t-6 <= -0.6: -0.005
> >
> > 78_t-0 <= -0.67: -0.005
> >
> > -0.81 < 90_t-7 <= 0: 0.005
> >
> > -0.68 < 30_t-7 <= 0: 0.005
> >
> > 136_t-5 <= -0.57: -0.005
> >
> > -0.01 < 115_t-3 <= 0.64: 0.005
> >
> > 96_t-1 > 0.6: 0.005
> >
> > 0 < 44_t-7 <= 0.89: -0.005
> >
> > 44_t-1 <= -0.57: 0.005
> >
> > -0.67 < 29_t-6 <= 0: 0.005
> >
> > -0.67 < 106_t-3 <= 0: 0.005
> >
> > -0 < 143_t-3 <= 0.62: 0.005
> >
> > 92_t-3 > 0.6: 0.005
> >
> > 86_t-4 <= -0.6: -0.005
> >
> > -0 < 69_t-0 <= 0.67: 0.005
> >
> > 39_t-6 > 0.65: 0.005
> >
> > 93_t-6 > 0.57: -0.005
> >
> > 89_t-4 > 0.59: -0.005
> >
> > 150_t-4 <= -0.63: 0.005
> >
> > -0 < 89_t-2 <= 0.59: 0.005
> >
> > 0.02 < 50_t-1 <= 0.59: 0.005
> >
> > -0.67 < 83_t-1 <= 0: -0.005
> >
> > 0 < 71_t-5 <= 0.69: 0.005
> >
> > 149_t-6 <= -0.64: 0.004
> >
> > 147_t-0 <= -0.72: 0.004
> >
> > -0.81 < 90_t-6 <= 0: 0.004
> >
> > -0.74 < 35_t-6 <= -0: 0.004
> >
> > 71_t-7 <= -0.64: -0.004
> >
> > -0.58 < 24_t-0 <= -0.02: -0.004
> >
> > -0.01 < 114_t-0 <= 0.64: -0.004
> >
> > -0.71 < 155_t-3 <= -0.01: -0.004
> >
> > -0.74 < 36_t-0 <= -0.01: 0.004
> >
> > 64_t-5 > 0.9: -0.004
> >
> > 85_t-6 <= -0.67: -0.003
> >
> > 0 < 93_t-4 <= 0.57: -0.003
> >
> > -0.57 < 136_t-4 <= 0: -0.003
> >
> > The top 5 features are:
> > 1. Count Positive-20: the average number of times the closing price was higher than the previous day's close over last 20 days.
> > 2. Volume Sum Difference-20: the difference between the number of volume increases and volume decreases over last 20 days.
> > 3. Volume Sum Positive-30: the proportion of volume increases over last 30 days.
> > 4. Standard deviation of volume-20: the standard deviation of volume over last 20 days.
> > 5. Highest price for the last day.
> >
> > Hope it addresses your concern. If there is any remaining concerns, please feel free to tell us and we are always more than happy to discuss.
> >
> > Best regards,
> >
> > Submission 8851 Authors

---

> > > ### Author Response · Authors · 2024-12-03
> > > **Greatly appreciate your support!**
> > >
> > > Dear Reviewer $\color{blue}\text{ZDKm}$,
> > >
> > > We are extremely grateful for the insightful suggestions to make our work better!
> > >
> > > We've tried our best to address your concerns. As the discussion phase comes to a close, if our response has solved most of your concerns, we humbly ask if you might consider raising your score.
> > >
> > > If you believe there are still areas in our paper that could be further improved, we would be more than happy to engage in any additional discussion to address any remaining concerns, particularly as the rebuttal period is about to conclude in just a few hours.
> > >
> > > Thank you once again for your thoughtful engagement! Your support at this stage is immensely important to us!
> > >
> > > Best regards,
> > >
> > > Submission 8851 Authors

---

### Author Response · Authors · 2024-11-25
**Gentle Reminder**

Dear reviewers,

Thanks again for your valuable time and insightful comments. We have provided thorough responses to each reviewer. As our work is a boardline paper, we sincerely hope you can look through the responses and update the scores if your concerns have been resolved. We are more than happy to further discuss if the concerns have not been fully addressed. Please feel free to let us know if you still have any questions.

Best regards!
Submission 8851 authors

---

### Author Response · Authors · 2024-12-04
**General Response**

Dear reviewers,

We truly appreciate your efforts in reviewing our paper. Constructive reviews has enhanced the quality of the paper. Here we would like to summarize the strengths of this work acknowledged by the reviewers, and the responses we have made to address all the reviewers' concerns.

---

### **Strengths**:
1. Motivation: The problem is attractive and lucrative (Reviewer vZrE); The concept is interesting (Reviewer BbWb); appropriate motivation provided by the authors (Reviewer 14Ku).
2. Presentation: Good writing and organization (Reviewer ZDKm); The paper is well-written (Reviewer 14Ku).
3. Novelty: The novel use of diffusion models is technically sound (Reviewer ZDKm).
4. Empirical Improvements: Well-designed Experiments and shows significant improvements in excess return (Reviewer ZDKm, Reviewer BbWb, Reviewer 14Ku).

---

### **Major concerns raised by reviewers**:
1. Generalization and overfitting (Reviewer ZDKm)
2. More baselines in stock market forecasting and data augmentation (Reviewer ZDKm, Reviewer 14Ku)
3. More test metric, explanation of RankIC and IC, significance test (Reviewer ZDKm, Reviewer BbWb, Reviewer 14Ku)
4. Some typos may hinder reading experience, a roadmap early on could be helpful. (Reviewer BbWb)

---

### **Response and Revision from us**:
1. - For the generalization issue, our target domain is within the source domain the characteristics of the two datasets will not be fundamentally different. We've added an explanation in (Line 244 ~ Line 249) to avoid misunderstanding.
    - For the overfitting issue, we propose a novel loss-guided diffusion mechanism which introduces stronger noise to easily fitted data points, information ratio results from Figure 7 shows the effectiveness of this mechanism in alleviating overfitting and decreasing volatility.

2. We've added MTMD, the SOTA stock market forecasting method in Qlib as our baseline to Table 1, 2 and 8. Furthermore, we've added Quant-GAN, COSCI-GAN, and TimeVAE as additional multivariate time-series sequence generation baselines in Figure 8. We also add significance test and the model performance comparison in the best case.

3. We've added MSE and RMSE as our test metric to further show the predict ability of DiffsFormer. The relationship of IC and RankIC for high-volatility and low-volatility stocks are further explained, which is consistent with the observation in MSE and RMSE figure.

4. We've fixed the typos and polish the sentences. We also add a roadmap near the end of the introduction.

---

Thanks again for your efforts and time!

Best regards,

Submission 8851 Authors

---

### Meta-Review · Area_Chair_2nN3 · 2024-12-20

**Metareview:**

This paper introduced a diffusion model for data augmentation in financial markets for the problem of low signal-to-noise ratio. The authors showed that the proposed approach improves the performance of various models. This paper received mostly negative comments, and the major concern is lack of novelty, since some well-known methods are not discussed carefully. Besides, the quality of writing may not meet the bar. Thus, I recommended a rejection.

**Additional Comments On Reviewer Discussion:**

Though authors wrote a long rebuttal, the reviewers didn't think their concerns were fully addressed.

---

### Decision · Program_Chairs · 2025-01-22

Reject